# Mitigating Information Impedance in Deep Spiking Neural Networks via Multi-Stage Foundation-Model Distillation

## Abstract

Brain-inspired spiking neural networks (SNNs) hold great promise for low-power, event-driven computation. Yet, their performance is fundamentally constrained by information impedance induced by spiking activations and spike-based propagation, a challenge that becomes more severe in deeper architectures and under limited time-steps. In this work, we conduct an information-theoretic analysis to reveal that such information impedance constitutes a key bottleneck to the learning of deep SNNs. To address it, we propose a multi-stage knowledge distillation (KD) method that leverages a high-capacity teacher model (DINOv2) to enhance the information extraction and transmission capabilities of SNNs. By decomposing a deep high-impedance path into low-impedance stages, our method effectively mitigates the representational bottlenecks caused by spike quantization. Extensive experiments demonstrate that our method substantially boost the learning of deep residual SNNs, e.g., on ImageNet-1K with ResNet-101, our method achieves 77.14% top-1 accuracy, which surpasses the prior SOTA by 2.93%. The gains are particularly significant for fully-spiking SNNs and deeper models. Importantly, while vanilla KD has been shown sufficient for ANNs on large-scale datasets, we show that for SNNs it is far from sufficient, and overcoming the information impedance is essential to fully unlock the potential of SNN distillation. Code is available at `https://anonymous.4open.science/r/MSKD-SNN-84AE`.

## 1 Introduction

Emulating biological neural networks, spiking neural networks (SNNs) have been recognized for their potential in energy-efficient, event-driven computing (Maass, 1997; Zenke & Ganguli, 2018; Lee et al., 2016; Neftci et al., 2019). In recent years, SNNs have attracted considerable research attention and achieved much progress (Fang et al., 2021; Meng et al., 2023; Hu et al., 2021; Li et al., 2021a; Yin et al., 2021; Xiao et al., 2022; Fang et al., 2023; Zhou et al., 2022; Yao et al., 2024). However, due to the information loss introduced by the discretization inherent to spiking neuron models, training deep fully-spiking networks (e.g., spiking ResNet-50/101) remains challenging. This problem becomes more pronounced in deeper networks and under restricted inference time-steps. Increasing the number of time-steps can partially mitigate the issue, but at the cost of higher energy consumption and latency, and the performance remains fundamentally limited.

Despite the progress in the past few years, training deep fully-spiking networks (e.g., spiking ResNet-50/101) is still a difficult problem. To address this challenge, hybrid-SNN architectures have been proposed by incorporating non-spiking operations (Fang et al., 2021; Hu et al., 2024; Zhou et al., 2024). Although these approaches achieve notable improvement in training performance, the introduction of non-spiking operations incurs additional computational overhead, which compromises the energy efficiency of SNNs, and may be infeasible on neuromorphic hardware that only supports spike-based operations. To enhance the training of deep fully-spiking SNNs, the dual-stream training (DST) method adopts a detachable auxiliary accumulation pathway to boost the training performance of spiking residual networks (Chen et al., 2023). Another line of research focuses on improving SNN training via knowledge distillation (KD), where an ANN teacher is used to help overcome the inherent difficulties of SNN training. A number of KD methods have been

developed (Yu et al., 2025c;b; Guo et al., 2024; Xu et al., 2023; Hong et al., 2023; Dong et al., 2024). However, most of these methods have not been evaluated on deep networks such as spiking ResNet-50/101, thus their effectiveness on deep SNNs is unclear.

Although numerous heuristic approaches have been developed to alleviate the training difficulties of deep SNNs, a rigorous analysis of the underlying causes remains absent. In this work, we revisit this challenge from an information-theoretic perspective and provide new insights that address the problem at its core. Our theoretical and empirical analyses reveal that deeper SNNs suffer from increasing information impedance, induced by spiking activations and spike-based propagation. This information impedance emerges as a fundamental bottleneck to the training of deep SNNs. Based on this insight, we propose a multi-stage KD method that leverages a high-capacity teacher to enhance information flow and improve the training of deep SNN. The main contributions of this work are as follows:

- We provide an in-depth analysis of the underlying challenges of deep SNNs training from an information-theoretic perspective. We show both theoretically and empirically that spiking activations and spike-based propagation lead to higher information impedance as depth increases. This information impedance fundamentally limits the training performance of deep SNNs.

- We propose a multi-stage KD method, which decomposes a high-impedance path into multiple low-impedance stages while using a high-capacity (informative) teacher to strengthen information transmission. This effectively mitigates the representational bottlenecks caused by spike quantization.

- We conduct extensive experiments in comparison with SOTA direct training and distillation methods, which demonstrate that our method substantially improve the learning of deep residual SNNs. On ImageNet-1K with ResNet-101, it achieves 77.14% top-1 accuracy, which surpasses the previous best by 2.93%. The gains are particularly significant for deeper and fully-spiking SNNs.

Notably, in ANNs, prior work has shown that vanilla KD is sufficient on large-scale datasets such as ImageNet-1K (Hao et al., 2023). This raises a natural question: *Is vanilla KD also sufficient for SNNs on large-scale datasets?* Our findings provide a clear negative answer. Unlike ANNs, vanilla KD is far from sufficient for SNNs. The intrinsic information impedance of SNNs fundamentally hinders direct KD transfer. Our results demonstrate that explicitly addressing information impedance is essential for unlocking the full potential of SNN distillation on large-scale datasets. By overcoming this information impedance, we show that deep SNNs can achieve substantially improved performance and narrow the gap with their ANN counterparts.

## 2 RELATED WORK

**SNN Learning Methods.** The bio-plausibility and energy-efficiency of SNNs has inspired much research in the neuromorphic computing community (Maass, 1997; Zenke & Ganguli, 2018; Lee et al., 2016; Rueckauer et al., 2017; Wu et al., 2018; Neftci et al., 2019; Lee et al., 2020; Fang et al., 2021; Radhakrishnan et al., 2021; Zheng et al., 2021; Han et al., 2020; Rathi & Roy, 2021). Many early works focus on ANN-to-SNN conversion, which transfers pre-trained artificial neural networks to the spiking domain to achieve high accuracy on deep SNN architectures (Hu et al., 2023; Wu et al., 2021; Pérez-Carrasco et al., 2013; Han et al., 2020; Bu et al., 2023). More recently, direct training with surrogate gradients and backpropagation-through-time (BPTT) has closed much of the performance gap with ANNs (Meng et al., 2023; Hu et al., 2021; Li et al., 2021a; Yin et al., 2021; Xiao et al., 2022; Fang et al., 2023; Zhou et al., 2022; Yao et al., 2024; Zhou et al., 2024). Meanwhile, approximate BPTT training algorithms have been introduced to mitigate BPTT's high memory and computational costs (Bellec et al., 2020; Bohnstingl et al., 2022; Xiao et al., 2022; Meng et al., 2023; Yin et al., 2023).

**Knowledge Distillation for SNNs.** Knowledge distillation (KD) (Hinton et al., 2015) has been widely adopted to transfer knowledge from high-capacity networks to compact models. For SNNs, early studies explored distilling from large isomorphic SNNs or pre-trained ANNs to smaller SNNs, which typically use logit matching or feature alignment (Kushawaha et al., 2021; Lee et al., 2021; Takuya et al., 2021; Guo et al., 2023a; Zhang et al., 2023). Representative methods include KD-SNN (Xu et al., 2023), Joint A-SNN (Guo et al., 2023a), SM (Deng et al., 2023), BKDSNN (Xu

et al., 2024), TSSD (Zuo et al., 2024), TKS (Dong et al., 2024), EnOF (Guo et al., 2024), and Super-SNN (Zhang et al., 2024). These methods employ a range of strategies, from combined logit-feature supervision to temporal-structural constraints, which have demonstrated notable gains on datasets such as CIFAR and ImageNet variants. However, as shown in our comparison in Table 1, most existing methods treat SNNs analogously to ANNs and overlook their distinct information dynamics induced by spiking activation and spike-based propagation.

To address this, we analyze the training challenge of deep SNNs from an information-theoretic perspective and propose a distillation method explicitly designed to overcome the high information impedance inherent in deep SNNs. Furthermore, while prior distillation methods are typically evaluated only on small to moderate-scale SNNs (e.g., spiking ResNet-18/34), our method has been extensively validated on deeper architectures such as spiking ResNet-50/101, where it demonstrates particular effectiveness.

**Multi-Stage Distillation.** Our multi-stage KD method resembles deep supervision (Lee et al., 2015; Zhang et al., 2019) and KD with auxiliary branches (Chen et al., 2021; Yu et al., 2024b) in form. Deep supervision (Lee et al., 2015) adds intermediate CE loss with one-hot labels mainly for optimization/gradient shortcutting. Self-distillation (Zhang et al., 2019) divides a network into several stages and distills knowledge within network itself. (Chen et al., 2021; Yu et al., 2024b) also consider stage separation and use auxiliary branches for cross-stage distillation or better feature alignment. In contrast, our intermediate heads distill high-entropy teacher posteriors from a foundation model, particularly targeting information impedance of deep SNNs. Importantly, our results show that (Table 3): *1)* For ANN, vanilla KD is sufficient and multi-stage KD does not yield any improvement, which is consistent with previous results (Hao et al., 2023). *2)* For both fully-spiking and hybrid SNNs, our method achieves significant improvement over vanilla KD, particularly for deeper and full-spiking models. Our contribution lies not only in architectural design but in a theory-driven motivation and mechanism specifically tailored for deep SNNs, whose effectiveness is strongly and consistently validated by our extensive results on deep SNNs.

## 3 PRELIMINARIES

Emulating biological neural network, SNNs process information via discrete, temporally sparse spike events, enabling energy-efficient computation on neuromorphic hardware. A widely used spiking neuron model is the leaky integrate-and-fire (LIF) neuron model (Abbott, 1999; Burkitt, 2006). Let $u_l^t$ denote the membrane potential of layer $l$ at time-step $t$, and $z_l^t \in \{0, 1\}$ the spike output, the discrete-time LIF dynamics can be expressed as

$$u_l^t = \alpha\, u_l^{t-1}(1 - z_l^{t-1}) + W_l z_{l-1}^t + b_l, \tag{1}$$

$$z_l^t = H(u_l^t - V_{th}), \tag{2}$$

where $\alpha \in (0, 1)$ is the membrane decay factor, $W_l$ and $b_l$ are the synaptic weights and bias, $V_{th}$ is the firing threshold, and $H(\cdot)$ is the Heaviside step function. A spike is generated once the membrane potential crosses the threshold $V_{th}$, after which it is reset via the term $(1 - z_l^{t-1})$. Since $H(\cdot)$ is non-differentiable, surrogate gradient methods (Neftci et al., 2019) are typically used to enable gradient-based training of deep SNNs via backpropagation through time (BPTT).

## 4 INFORMATION FLOW ANALYSIS IN SNNS

Let $X$ and $Y$ denote the data and label variables, respectively. Consider an SNN with $L$ layers, where the spiking representation of the $l$-th layer is denoted as $Z_l \in \{0, 1\}^{n_l}$ with $n_l$ neurons. The network induces the Markov chain $Y \to X \to Z_1 \to Z_2 \to \cdots \to Z_L$. From the perspective of information theory, the ideal learning objective can be formulated as extracting task-relevant information about $Y$ from the input $X$ while discarding nuisance variability (Shwartz-Ziv & Tishby, 2017; Achille & Soatto, 2018; Saxe et al., 2019):

$$\max I(Y; Z_L) \quad \text{subject to} \quad I(X; Z_L|Y) = 0, \tag{3}$$

where $I(\cdot; \cdot)$ denotes the mutual information. The criterion $\max I(Y; Z_L)$ is equivalent to minimizing the conditional entropy $H(Y|Z_L)$, which measures the uncertainty of predicting $Y$ given the representation $Z_L$. Intuitively, the desired representation should retain all task-relevant information while eliminating irrelevant redundancy. In the ideal case, the representation $Z_L$ satisfies both *sufficiency* ($I(Y; Z_L) = I(Y; X)$) and *minimality* ($I(X; Z_L|Y) = 0$), i.e., capturing only the information necessary for predicting $Y$.

### 4.1 CAPACITY OF SPIKING REPRESENTATION AND INFORMATION BOTTLENECK OF SNNS

Let layer $l$ have $n_l$ neurons observed over $T$ time-steps. Denote the neuron spikes of layer $l$ at the $t$-th time-step by $Z_l^t := [z_{l,1}^t, \cdots, z_{l,n_l}^t] \in \{0,1\}^{n_l}$, and $Z_l^{1:T} := [Z_l^1, \cdots, Z_l^T] \in \{0,1\}^{n_l \times T}$ collects the full spike trains of layer $l$. Suppose that the global firing-rate is constrained by some $\rho_l \in (0,1)$ as $\frac{1}{n_l T} \sum_{i=1}^{n_l} \sum_{t=1}^{T} \mathbb{E}[z_{l,i}^t] \le \rho_l$, where the expectation is taken over data distribution. Here, we consider the spike-count coding, which is widely used in deep SNNs, and denote the layer count representation by $C_l = [c_{l,1}, \cdots, c_{l,n_l}]$, where $c_{l,i} = \sum_{t=1}^{T} z_{l,i}^t \in \{0, \ldots, T\}$.

**Proposition 1 (Representational entropy capacity of an SNN layer)** *Under spike-count coding, the entropy capacity of the $l$-th layer under $T$ time-steps is*

$$\mathcal{C}(C_l) = \sup H(C_l) = n_l H_{\mathrm{Bin}}(T, \tilde{\rho}_l) \le n_l \log(T+1), \tag{4}$$

*where $H_{\mathrm{Bin}}(T, p) = -\sum_{k=0}^{T} \binom{T}{k} p^k (1-p)^{T-k} \log\left(\binom{T}{k} p^k (1-p)^{T-k}\right)$ is the entropy of Binomial distribution, and $\tilde{\rho}_l = \min\{\rho_l, \frac{1}{2}\}$. As $T \to \infty$, the entropy capacity can be expressed as*

$$\mathcal{C}(C_l) = \frac{n_l}{2} \log\left(2\pi e T \tilde{\rho}_l (1 - \tilde{\rho}_l)\right) + O\left(\frac{1}{T}\right) \le n_l \log(T+1). \tag{5}$$

Proposition 1 characterizes the representational entropy capacity of a single SNN layer, which depends jointly on the number of neurons $n_l$ and the firing rate constraint $\rho_l$. Under spike-count coding, this capacity scales only logarithmically with the number of time-steps $T$.

**Proposition 2 (Layer-wise bottleneck for SNNs)** *Assume a forward Markov chain $X \to Z_1^{1:T} \to \cdots \to Z_L^{1:T}$. For spike-count coding at the output layer, it holds that $I(X; C_L) \le H(C_L) \le n_L H_{\mathrm{Bin}}(T, \tilde{\rho}_L) \le n_L \log(T+1)$. As $T \to \infty$, the entropy of Binomial distribution can be expressed as $H_{\mathrm{Bin}}(T, p) = \frac{1}{2} \log(2\pi e T p(1-p)) + O(\frac{1}{T})$, and it holds that*

$$I(X; C_L) \le \frac{n_L}{2} \log\left(2\pi e T \tilde{\rho}_L (1 - \tilde{\rho}_L)\right) + O(\tfrac{1}{T}). \tag{6}$$

*Moreover, assume a count-sufficiency condition of $X \to C_1 \to \cdots \to C_L$, i.e., $I(Z_l^{1:T}; C_L|C_l) = 0$, then for any $l \le L$ it follows that $I(X; C_L) \le I(X; C_l) \le H(C_l) \le n_l H_{\mathrm{Bin}}(T, \tilde{\rho}_l)$ and*

$$I(X; C_L) \le \min_{1 \le l \le L} n_l H_{\mathrm{Bin}}(T, \tilde{\rho}_l). \tag{7}$$

Proposition 2 shows the layer-wise information bottleneck in SNNs. Under spike-count coding, the mutual information between the input $X$ and the output counts $C_L$ is limited by the smallest layer capacity along the feedforward chain. In particular, deeper models introduce additional potential bottlenecks, since attenuation of firing rates across layers tends to occur and further tightens the entropy bound. This result reveals the intrinsic difficulty of reliable information transmission in deep SNNs.

Prior works (Wang et al., 2021; Yu et al., 2025a) have applied the information bottleneck principle to analyze deep ANNs in the context of locally supervised (block-wise) learning, which focuses on how local objectives shape representation compression and task-relevant information. In contrast, we study the information flow in deep SNNs and identify a unique information impedance caused by spike-based computation, e.g., spike quantization, limited time-steps, and spike-based propagation, which leads to depth-accumulated information attenuation.

### 4.2 EMPIRICAL EVALUATION OF INFORMATION FLOW

We empirically examine the information dynamics in the data process $X \to Z_1 \to Z_2 \to \cdots \to Z_L$ of SNNs by estimating the mutual information $I(X; Z_l)$. Let $E(X|Z_l)$ denote the expected

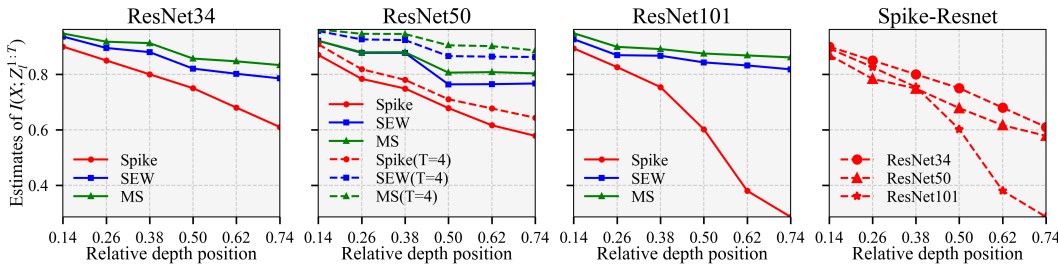

Figure 2: Estimation of $I(X; Z_l)$ across network depth for three deep SNN architectures: a fully-spiking SNN Spike-ResNet and two hybrid SNNs (SEW-ResNet and MS-ResNet), each evaluated at ResNet-34/50/101 scales.

reconstruction error of $X$ from the representation $Z_l$. Since $I(X; Z_l) = H(X) - H(X|Z_l) \geq H(X) - E(X|Z_l)$ (Vincent et al., 2008; Hjelm et al., 2017), we approximate the mutual information through auxiliary reconstruction networks that minimize the reconstruction distortion of the input $X$, i.e., $I(X; Z_l) \approx \max_{\Theta} \big( H(X) - E(X|D_{\Theta}(Z_l)) \big)$, where $D_{\Theta}(\cdot)$ denotes a reconstruction network parameterized by $\Theta$.

We systematically analyze the effect of three factors, architecture, depth and time-steps, on the information propagation of spiking ResNets. **Model architecture:** We consider three deep SNN architectures, including a fully-spiking SNN and two hybrid-SNNs, as shown in Figure 1. **Model depth:** We evaluate three different model scales ResNet-34/50/101. By normalizing the layer index to $[0, 1]$, it enables a fair comparison of information trajectories across networks of different depths. **Time-steps:** We further examine the effect of the number of time-steps on the information transmission capability of spiking models, e.g., spiking ResNet50 with $T \in \{1, 4\}$.

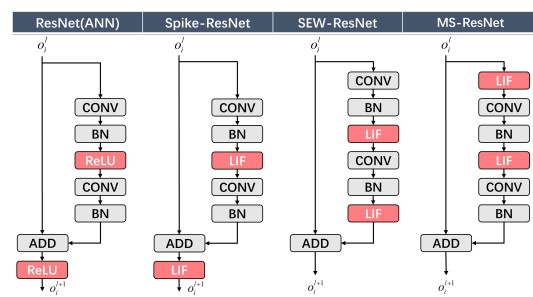

Figure 1: Block structure of three SNN ResNet architectures.

The results, as shown in Figure 2, reveal the following observations:

- **Information loss in fully-spiking SNNs.** The information flow $I(X; Z_l)$ in fully-spiking SNNs exhibits a much sharper decay compared to hybrid-SNNs, especially for deeper models such as ResNet-101. This explains the empirical performance gap observed between fully-spiking and hybrid SNN architectures.

- **Depth aggravates bottlenecks.** The information gap between fully-spiking SNNs and their counterparts becomes more pronounced as depth increases. This aligns with the above theoretical analysis: deeper spiking architectures introduce more layers of spike-based transformations, which exacerbates the attenuation of information across the network.

- **Mitigation via time-steps.** Increasing the number of time-steps can alleviate the information loss in both fully-spiking and hybrid SNNs. This is consistent with the intuition that multi-step spike encoding provides richer temporal representations, which can partially compensate for the reduced representational capacity of single time-step.

The above findings indicate that the intrinsic spiking mechanism of fully-spiking SNNs imposes substantially higher impedance on information flow compared to hybrid-SNNs. This fundamental limitation explains the training inefficiency of fully-spiking SNNs, as information critical for learning is progressively attenuated across layers. Furthermore, the limited information capacity exacerbates performance degradation under small or single time-steps. Finally, the exacerbated information loss in deeper architectures makes the training of deep fully-spiking SNNs particularly challenging.

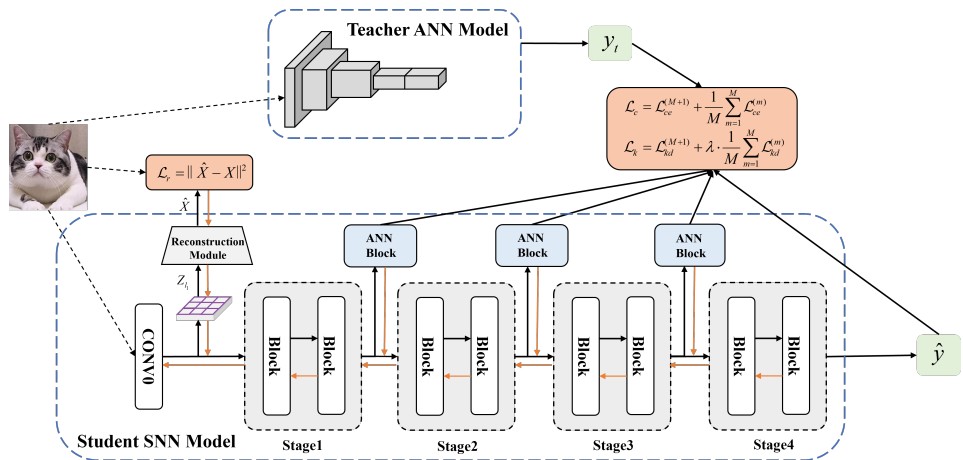

Figure 3: Illustration of the proposed multi-stage distillation method.

# 5 ENHANCING DEEP SNNs LEARNING VIA MULTI-STAGE FOUNDATION-MODEL DISTILLATION

The above analysis shows that deep SNNs suffer from high information impedance that each layer has a finite entropy capacity, and the end-to-end mutual information is clamped by the smallest layer capacity, especially for deeper models and under small time-steps. To overcome the information impedance in deep SNNs, we propose to distill a strong foundation model (DINOv2) into SNNs at multiple stages via auxiliary ANN adapters, as shown in Figure 3. This design raises the effective information carried by intermediate SNN representations and eases the training difficulty. The rationale rests on three complementary pillars:

(1) **Foundation models provide more informative supervision.** While one-hot labels provide a low-entropy target, teacher posteriors $q_t(y|x)$ are soft high-entropy distributions that encode relative similarities (dark knowledge) (Hinton et al., 2015). Minimizing the loss $KL\big(q_t(\cdot|x)\|p_t(\cdot|Z_l)\big)$ injects more informative supervision at layer $l$, which effectively maximizes agreement with an estimate of the Bayes posterior. With a calibrated teacher, this increases the task-relevant information $I(Z_l; Y)$ and yields better-conditioned gradients than using one-hot targets.

(2) **Decomposing a deep high-impedance path into low-impedance stages.** Information impedance exacerbates with depth. The proposed multi-stage distillation places local objectives $KL(q_t\|p_t(\cdot|Z_l))$ at several checkpoints (intermediate layers), which supplies side-channel supervision close to where information is lost. This has two effects. *i)* Capacity matching per stage: Each block is encouraged to saturate its own entropy budget with task-relevant content, which raises $I(Z_l; Y)$ before further attenuation. *ii)* Optimization relief: Auxiliary losses shorten gradient paths and mitigate surrogate-gradient noise, which turns a long lossy pipeline into several easier subproblems. Empirically, this raises the measured information $I(X; Z_l)$.

(3) **Preserving input information with reconstruction loss.** To further mitigate the severe information loss caused by spiking encoding, we introduce a reconstruction module that enforces the recovery of the input $X$ from intermediate representations. The reconstruction loss $\mathcal{L}_{rec}(\hat{X}, X)$ explicitly increases the mutual information $I(X; Z_l)$, and encourages lower-layer SNN representations to retain more information about the input.

Specifically, for an SNN with $L$ sequential blocks with the representation after the $l$-th block being $Z_l$, to inject supervision at multiple stages, we attach $M$ auxiliary classification heads (intermediate heads) at selected depths $\{l_1, \ldots, l_M\} \subseteq \{1, \ldots, L\}$. Each head is implemented as a lightweight ANN adapter that maps the spiking feature $Z_{l_m}$ into logits $h_m : Z_{l_m} \mapsto \hat{y}_m$ for $m = 1, \cdots, M$, where $\hat{y}_m \in \mathbb{R}^C$ is the student prediction over $C$ classes at the $m$-th head. The final head $h_{M+1}$ corresponds to the output of the last block. For each head, we compute a standard cross-entropy loss with respect to the ground-truth label $y$: $\mathcal{L}_{ce}^{(m)} = -\sum_{c=1}^{C} \mathbb{I}(y = c) \log \sigma_c(\hat{y}_m)$, where $\sigma$ denotes the

softmax. The overall classification objective combines the auxiliary heads with the final head

$$\mathcal{L}_c = \mathcal{L}_{ce}^{(M+1)} + \sum_{m=1}^{M} w_c^{(m)} \mathcal{L}_{ce}^{(m)}, \tag{8}$$

where $w_c^{(m)}$ is a weighting coefficient for the cross-entropy loss of the $m$-th auxiliary head. We distill knowledge from a foundation model by aligning the softened logits of it and student at each head. For head $m$, the distillation loss is defined as

$$\mathcal{L}_{kd}^{(m)} = KL\left(\sigma(\hat{y}_m/\tau_m) \,\middle\|\, \sigma\left(y_t/\tau_m\right)\right), \tag{9}$$

where $\tau_m$ is the temperature. The multi-stage distillation objective aggregates across all heads as

$$\mathcal{L}_k = \mathcal{L}_{kd}^{(M+1)} + \sum_{m=1}^{M} w_k^{(m)} \mathcal{L}_{kd}^{(m)}, \tag{10}$$

where $w_k^{(m)}$ is a weighting coefficient for KL distillation loss of the $m$-th auxiliary head.

To preserve input information after spike encoding, we attach a lightweight reconstruction module $g$ after the first convolution layer $\hat{X} = g(Z_{l_1})$ and minimize the mean-squared error with the input $\mathcal{L}_r = \|\hat{X} - X\|^2$. The final training loss combines all three components as

$$\mathcal{L} = (1 - \alpha) \cdot \mathcal{L}_c + \alpha \cdot \mathcal{L}_k + \beta \cdot \mathcal{L}_r, \tag{11}$$

where $\alpha, \beta$ control the trade-off among supervised learning, multi-stage distillation, and input information preservation. This design provides both global and local supervision, improves optimization stability, and alleviates the information bottleneck in deep SNNs.

Compared with the vanilla distillation method, the proposed multi-stage foundation-model distillation method *(1)* supplies more informative targets by using a foundation-model as the teacher; *(2)* decomposes a deep high-impedance path into low-impedance stages and promotes each stage to saturate its own entropy budget with task-relevant content; *(3)* alleviates the severe information loss caused by spiking encoding using a reconstruction loss. These factors jointly enhance the information flow in deep SNNs and finally improve training accuracy.

## 6 EXPERIMENTS

We evaluate the effectiveness of the proposed method in comparison with SOTA direct training and distillation methods on typical classification and fine-grained classification tasks. We use BPTT with a sigmoid-based surrogate gradient $h(x, \alpha) = \frac{1}{1+e^{-\alpha x}}$ with $\alpha = 4$, implemented in SpikingJelly (Fang et al., 2020). We use DINOv2-B as the teacher. For CIFAR-10 and CIFAR-100, training is performed on a single V100 GPU, while ImageNet experiments use 8 V100 GPUs. We use the SGD optimizer with a momentum of 0.9 and a cosine annealing learning rate schedule for all datasets. Detailed hyperparameters are provided in Table 6 in Appendix.

### 6.1 RESULTS ON IMAGENET

On ImageNet-1K, we evaluate two representative families of deep spiking architectures: a fully-spiking variant Spike-ResNet and the SOTA hybrid design MS-ResNet. For both architectures, we instantiate three scales corresponding to ResNet-34/50/101 backbones, and consider time-steps $T \in \{1, 4\}$. The teacher DINOv2-B achieves an accuracy of 84.41%. As shown in Table 1, the proposed method substantially outperforms prior direct training and distillation baselines. It establishes new SOTA results for deep spiking ResNet on ImageNet-1K. The gains are more significant for deeper architectures. For instance, our method with MS-ResNet-101 ($T = 4$) achieves an improvement of 2.93% over the previous best MS-ResNet-104 ($T = 5$) reported in (Hu et al., 2024). Remarkably, MS-ResNet-101 with $T = 4$ reaches 77.14% Top-1 accuracy, which marks **the first spiking ResNet surpassing the 77% threshold on ImageNet-1K**.

The analysis in Section 4 reveals that fully-spiking architectures face more severe information impedance, which make the training of Spike-ResNet-50/101 particularly challenging. For example, direct training of Spike-ResNet-50 and Spike-ResNet-101 with $T = 4$ yields only 57.66%

Table 1: Top-1 accuracy (%) on ImageNet-1K with single-crop evaluation. (↑) denotes the improvement over the previous SOTA with the same architecture and time-step, except SEW-ResNet-101 ($T$=4) and MS-ResNet-101 ($T$=4) are compared with MS-ResNet-104 ($T$=5) in (Hu et al., 2024).

| Method | | Model | Time step ($T$) | Acc. (%) |
|---|---|---|---|---|
| | Spike (Fang et al., 2021) | Spike-ResNet-34 | 4 | 61.86 |
| | | Spike-ResNet-50 | 4 | 57.66 |
| | | Spike-ResNet-101 | 4 | 31.79 |
| | STBP-tdBN (Zheng et al., 2021) | Spike-ResNet-34 | 6 | 63.72 |
| | | Spike-ResNet-50 | 6 | 64.88 |
| | GLIF (Yao et al., 2022) | Spike-ResNet-34 | 4 | 67.52 |
| | MPBN (Guo et al., 2023b) | Spike-ResNet-34 | 4 | 64.71 |
| Direct | TET (Deng et al., 2022) | Spike-ResNet-34 | 6 | 64.79 |
| training | SEW ResNet (Fang et al., 2021) | SEW-ResNet-34 | 4 | 67.04 |
| | | SEW-ResNet-50 | 4 | 67.78 |
| | | SEW-ResNet-101 | 4 | 68.76 |
| | MS ResNet (Hu et al., 2024) | MS-ResNet-34 | 6 | 69.42 |
| | | MS-ResNet-104 | 5 | 74.21 |
| | DST (Chen et al., 2023) | Spike-ResNet-34 | 4 | 66.45 |
| | | Spike-ResNet-50 | 4 | 67.69 |
| | | Spike-ResNet-101 | 4 | 68.38 |
| | SM (Deng et al., 2023) | Spike-ResNet-34 | 4 / 6 | 68.25 / 69.35 |
| | RateBP (Yu et al., 2024a) | SEW-ResNet-34 | 4 | 65.84 |
| | | MS-ResNet-34 | 4 | 70.01 |
| | ETC (Zhao et al., 2025) | SEW-ResNet-34 | 4 / 6 | 68.54 / 69.64 |
| | KDSNN (Xu et al., 2023) | SEW-ResNet-34 | 4 | 67.18 |
| | LaSNN (Hong et al., 2023) | SEW-ResNet-34 | 4 | 66.94 |
| | TKS (Dong et al., 2024) | SEW-ResNet-34 | 4 | 69.60 |
| With | EnOF (Guo et al., 2024) | SEW-ResNet-34 | 4 | 67.40 |
| distillation | FRTD (Yu et al., 2025b) | MS-ResNet-34 | 1 / 4 | 66.50 / 71.04 |
| | TSER (Yu et al., 2025c) | SEW-ResNet-34 | 4 | 73.16 |
| | | Spike-ResNet-34 | 1 / 4 | **61.74 / 69.34** (↑1.09) |
| | | Spike-ResNet-50 | 1 / 4 | **66.35 / 72.08** (↑4.39) |
| | | Spike-ResNet-101 | 1 / 4 | **67.00 / 72.86** (↑4.48) |
| | | SEW-ResNet-34 | 1 / 4 | **68.05 / 73.25** (↑0.09) |
| | Ours | SEW-ResNet-50 | 1 / 4 | **69.64 / 74.58** (↑0.82) |
| | | SEW-ResNet-101 | 1 / 4 | **71.91 / 76.77** (↑2.56) |
| | | MS-ResNet-34 | 1 / 4 | **68.44 / 73.46** (↑0.30) |
| | | MS-ResNet-50 | 1 / 4 | **70.08 / 74.85** (↑1.09) |
| | | MS-ResNet-101 | 1 / 4 | **72.67 / 77.14** (↑2.93) |

and 31.79% accuracy, respectively, substantially worse than Spike-ResNet-34. This highlights that advanced training strategies are indispensable for fully-spiking networks to reach competitive performance. While prior work, e.g., DST (Chen et al., 2023) using auxiliary accumulation pathways, has partially alleviated this issue, our method delivers further substantial gains, e.g., +4.39% on Spike-ResNet-50 and +4.48% on Spike-ResNet-101, reaching 72.08% and 72.86% Top-1 accuracy, respectively. Moreover, compared to direct training, the improvements are more significant, e.g., +14.42% and +41.07% on Spike-ResNet-50 and Spike-ResNet-101, respectively. These results establish **the first fully-spiking ResNets surpassing the 70% accuracy on ImageNet-1K.**

## 6.2 RESULTS ON FINE-GRAINED CLASSIFICATION AND CORRUPTION DATA

We further evaluate the performance of different teacher models on ImageNet-C and four fine-grained classification benchmarks. ImageNet-C contains 15 corruption types, each with five severity. We use MS-ResNet-34 pre-trained on ImageNet as the student model, and report the test accuracy

Table 2: Performance (MS-ResNet-34) on ImageNet-C and fine-grained datasets under different teachers (Top-1 accuracy %).

| Method | Teacher | Acc | ImageNet-C | | Fine-grained datasets | | | | |
|---|---|---|---|---|---|---|---|---|---|
| | Params | | Severity 5 | Severity 1-5 | Cars | Birds | Dogs | Aircraft | Avg. |
| w/o Teacher | – | 63.80% | 20.35 | 34.90 | 42.83 | 42.96 | 67.93 | 42.24 | 48.99 |
| TSER | 21.8M | 65.18% | 21.82 | 36.81 | 42.32 | 42.54 | 68.66 | 41.08 | 48.65 |
| *Our method with different teachers* | | | | | | | | | |
| ResNet34 | 21.8M | 65.33% | 21.66 | 36.52 | 41.71 | 42.61 | 68.76 | 41.21 | 48.57 |
| DINOv2-Small | 21M | 65.40% | 22.94 | 37.67 | 42.13 | 44.60 | 70.49 | 42.69 | 49.98 |
| DINOv2-Base | 86M | 65.95% | 23.40 | 38.40 | 42.57 | 45.03 | 70.22 | 42.88 | 50.18 |
| DINOv2-Large | 300M | 65.89% | 23.93 | 38.84 | 43.12 | 45.49 | 70.92 | 43.58 | 50.78 |

Table 3: Comparison of accuracy (%) on ImageNet100 across different architectures and depths. *Baseline* denotes the direct training method. $\mathcal{L}_{kd}^{(M+1)}$ denotes the vanilla distillation method using only the student's final output for distillation.

| Method | Spike-ResNet-34 | MS-ResNet-34 | SEW-ResNet-34 | ResNet34 (ANN) | Spike-ResNet-50 | Spike-ResNet-101 |
|---|---|---|---|---|---|---|
| Baseline | 71.50 | 76.78 | 75.82 | 82.58 | 69.85 | 9.78 |
| $\mathcal{L}_{kd}^{(M+1)}$ | 77.04 (↑5.54) | 79.22 (↑2.44) | 78.80 (↑2.98) | **87.76 (↑5.18)** | 77.36 (↑7.51) | 15.60 (↑5.82) |
| $w/o\ \mathcal{L}_r$ | 80.32 (↑8.82) | 82.14 (↑5.36) | 81.65 (↑5.83) | 87.60 (↑5.02) | 80.26 (↑10.41) | 68.76 (↑58.98) |
| Ours | **80.81 (↑9.31)** | **82.50 (↑5.72)** | **82.06 (↑6.24)** | 87.53 (↑4.95) | **80.86 (↑11.01)** | **69.36 (↑59.58)** |

on ImageNet-C. We use linear probes for the fine-grained datasets that train additional classifiers for each of them. As shown in Table 2, despite the fact that a large number of classes in those four fine-grained datasets do not correlate with ImageNet-1K, the proposed distillation method enhances the generalization and robustness, with the performance improves as the model size of the teacher increases.

### 6.3 Ablation Study

**Effectiveness of the proposed multi-Stage distillation.** We conduct ablation studies on ImageNet100 to examine the effectiveness of the proposed components. First, we consider a fixed depth of ResNet-34 and compare four architectures: fully-spiking ResNet, hybrid SEW-ResNet and MS-ResNet, and ANN ResNet. We then extend the evaluation to fully-spiking networks with different depths (ResNet-34/50/101) to examine the scalability.

The results in Table 3 show that, for ANN, the vanilla distillation method is sufficient and the proposed method does not yield further improvement. This is consistent with previous results that, for ANNs the vanilla distillation method is sufficient on large-scale datasets such as ImageNet-1K (Hao et al., 2023). In contrast, for both fully-spiking and hybrid SNNs, our method achieves significant improvement over the vanilla method. The improvement is particularly pronounced for the full-spiking ResNet, +9.31% over the direct training baseline.

The proposed method demonstrates particular effectiveness on deeper architectures. For Spike-ResNet-101, which suffers from pronounced information degradation under its fully-spiking design, our method achieves a remarkable improvement of +59.58% over direct training. These observations lead to two insights: *i)* the proposed multi-stage distillation substantially enhances the representational capacity of SNNs, with pronounced benefits on fully-spiking SNNs whose information flow is most constrained; *ii)* the advantages become more prominent as network depth increases, which indicates that our method enables effective training of large-scale fully-spiking models that were previously regarded as difficult to optimize. Overall, these results clearly demonstrate the effectiveness of our method in alleviating the information transmission bottleneck in deep SNNs.

**Impact of the auxiliary ANN adapters.** For efficiency, this part of ablation experiment only conducts 50 epochs training. Table 8 in Appendix shows the effect of the auxiliary ANN adapters, which shows that configurations with only one or two convolutional layers are sufficient for achieving satisfactory performance, whereas deeper adapters ($\geq 4$ layers) lead to a clear performance drop. Two layers design provides the best trade-off between accuracy gains and computational cost.

Table 9 in Appendix shows the effect of ANN adapter number. For shallower networks (e.g., ResNet-34), adding ANN adapters initially improves accuracy, but the performance gain saturates beyond

Table 4: Training cost comparison across different SNN ResNet on ImageNet1k (batch size=1024, DINOv2 denotes the DINOv2-base teacher, ResNet denotes the ANN ResNet teacher of the same depth).

| Methods | MS-ResNet-34 | | MS-ResNet-50 | | MS-ResNet-101 | |
|---|---|---|---|---|---|---|
| | Time (per epoch) | Mem (GB) | Time (per epoch) | Mem (GB) | Time (per epoch) | Mem (GB) |
| Direct Train | 6m52s | 78.7 | 10m45s | 167.5 | 14m53s | 228.1 |
| Vanilla KD (ResNet) | 7m38s (+11.2%) | 80.5 (+2.3%) | 11m29s (+6.8%) | 169.9 (+1.4%) | 16m34s (+11.3%) | 230.3 (+1.0%) |
| Vanilla KD (DINOv2) | 9m29s (+38.1%) | 89.8 (+14.1%) | 13m41s (+27.2%) | 173.4 (+3.5%) | 18m22s (+23.4%) | 235.4 (+3.2%) |
| Our Method (DINOv2) | 10m08s (+47.6%) | 91.0 (+15.6%) | 14m23s (+33.7%) | 176.4 (+5.3%) | 19m10s (+28.8%) | 240.4 (+5.4%) |

three adapters. In contrast, deep networks (e.g., ResNet-101) consistently benefit from additional ANN adapters, with substantially higher improvements than shallow counterparts. Moreover, compared with hybrid architectures, our method is particularly effective in fully-spiking networks.

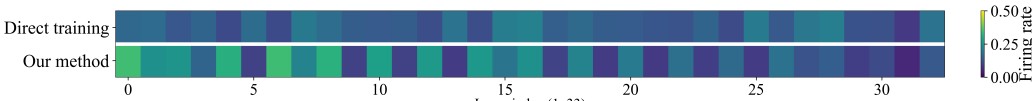

Figure 4: Layer-wise firing rate heatmaps for the direct training method and our method with Spike-ResNet-34 on ImageNet100.

**Training costs.** We compare the training overhead across four settings direct training, vanilla KD (ANN teacher), vanilla KD (DINOv2-base), and our method for different SNN ResNet students, as shown in Table 4. Compared with vanilla KD (DinoV2), our method introduces only a marginal increase in training time and memory usage. Using DINOv2-base as the teacher introduces an increase in training time. However, the relative overhead consistently decreases as the network depth grows.

**Firing rate distribution analysis.** As shown in the Figure 4, our method achieves superior performance with only a relatively small increase in firing rate (our method's average firing rate is 18.09% vs. 15.65% of direct training). This mitigates the progressive loss of activations in deep layers during direct training and endows more reliable information propagation throughout the architecture. Furthermore, early layers are more strongly activated to enhance feature extraction, while intermediate and deeper layers maintain moderate sparsity. This layer-by-layer distribution reflects a more balanced feature representation, which enables the model to achieve improved accuracy.

## 7 CONCLUSION

In this work, we investigated the fundamental challenge of training deep SNNs through the lens of information theory. Our analysis revealed that spiking activations and spike-based propagation introduce increasing information impedance as network depth increases, which constitutes a key bottleneck to effective training. To address this limitation, we proposed a multi-stage distillation method that decomposes high-impedance learning paths into low-impedance stages and leverages a high-capacity teacher to enhance information flow. Extensive experiments demonstrated that our method substantially improves the training of deep SNNs, which achieves new SOTA results on spiking ResNet-34/50/101. The performance gains are particularly pronounced for fully-spiking SNNs and deeper SNN models. Our method demonstrated the importance of explicitly addressing the information impedance in large-scale SNN training. Our findings not only advance the performance of deep SNNs but also provide new insights for developing scalable neuromorphic learning algorithms.

## REPRODUCIBILITY STATEMENT

We provide the source code and configuration of our algorithm at `https://anonymous.4open.science/r/MSKD-SNN-84AE`, including the models used and instructions for training them. Implementation details and all proofs, along with their explanations and underlying assumptions, are provided in the Appendix.

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

# A DETAILED EXPERIMENT SETTINGS

## A.1 DATASET AND HYPERPARAMETERS

Table 5 provides an overview of the datasets employed in this study. CIFAR10 and CIFAR100 consist of 10 and 100 categories, respectively. ImageNet100 is a 100-class subset derived from ImageNet-1K, while the full ImageNet-1K dataset comprises 1,000 categories. To assess model robustness, we additionally utilize ImageNet-C, which applies 15 types of perturbations to the ImageNet validation set, each with five levels of severity. Table 6 shows the hyperparameters for training ImageNet-1k.

Table 5: Summary of the datasets

| Dataset | #Train | #Test | #Corr. | #Severity | #Class. |
|---|---|---|---|---|---|
| CIFAR10 | 50,000 | 10,000 | - | - | 10 |
| CIFAR100 | 50,000 | 10,000 | - | - | 100 |
| ImageNet100 | 127,000 | 5,000 | - | - | 100 |
| ImageNet | 1,281,167 | 50,000 | - | - | 1000 |
| ImageNet-C | - | 50,000 | 15 | 5 | 1000 |

Table 6: Training hyperparameters for MS-ResNet-34 and MS-ResNet-101 on ImageNet.

| Hyperparameter | MS-ResNet-34 | MS-ResNet-101 |
|---|---|---|
| Learning rate | 0.6 | 0.6 |
| Batch size | 2048 | 350 |
| Weight decay | $2 \times 10^{-5}$ | $2 \times 10^{-5}$ |
| Decay rate | 0.2 | 0.2 |
| Time steps | 1 | 4 |
| Epochs | 300 | 300 |
| Alpha | 0.2 | 0.2 |
| Temperature | 5.0 | 5.0 |
| Beta | 0.9 | 0.9 |

## A.2 AUXILIARY ANN ADAPTERS ARCHITECTURE

As shown in Table 7, a two-layer convolutional ANN Block is shown, along with a prediction head. The convolution kernel sizes of the two convolutional layers are 7×7 and 3×3, respectively. Each level includes convolution, batch normalization, and LeakyReLU to gradually extract and refine features. This is followed by a prediction head consisting of adaptive average pooling and a fully connected layer.

Our results show that the setting of $K = 128$ offers the most favorable trade-off, balancing accuracy improvements with parameter growth. Furthermore, we observe that Spike-ResNet-34 is considerably more sensitive to such channel variations, exhibiting larger fluctuations in accuracy, while MS-ResNet-34 remains relatively stable across different widths.

## A.3 HYPERPARAMETER SENSITIVITY EVALUATION

We further conduct a hyperparameter sensitivity study on ImageNet100 using MS-ResNet-34 as the backbone, as illustrated in Figure 6. The results show that both the scaling factor $\alpha$ and the temperature parameter achieve their best performance at intermediate values, while overly small or large settings lead to noticeable accuracy degradation, a trend consistent with prior observations in knowledge distillation literature. In contrast, the model exhibits a relatively robust response to

Table 7: Architecture of the auxiliary ANN adapters.

| Component | Layers |
|---|---|
| Conv block 1 | Conv2d ($64 \rightarrow 128$, $7 \times 7$, stride 2, padding 3)
BatchNorm2d (128)
LeakyReLU ($\alpha = 0.01$) |
| Conv block 2 | Conv2d ($128 \rightarrow 128$, $3 \times 3$, stride 1, padding 1)
BatchNorm2d (128)
LeakyReLU ($\alpha = 0.01$) |
| Prediction head | AdaptiveAvgPool2d ($2 \times 2$)
Flatten
Linear ($512 \rightarrow 512$)
BatchNorm1d (512), LeakyReLU ($\alpha = 0.01$)
Linear ($512 \rightarrow 1000$) |


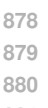
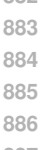
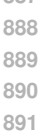

Figure 5: Comparison of different auxiliary ANN adapters channel widths with MS-ResNet-34 and Spike-ResNet-34 on ImageNet100.

variations in the weighting coefficient $\beta$, with only minor performance fluctuations and a slight preference for larger values. The model performs slightly better when the $\beta$ parameter takes a larger value. We further conduct three depth-aware ablations on Eqs. (8)–(10): (i) assigning depth-dependent weights to intermediate heads,(ii) placing denser ANN adapters toward later layers, and (iii) using a depth-dependent temperature. As shown in Tables 10, 11 and 12, moderately increasing the deep-layer $w_c$ and $w_k$ improves performance, whereas varying the temperature has little effect. Moreover, our results indicate that uniformly distributing the ANN adapters yields higher accuracy, while concentrating them in deeper layers degrades performance.

### A.4 ENERGY CONSUMPTION ANALYSIS

Based on the energy model in Kundu et al. (2021), our energy analysis is summarized in Table 13. As shown in Table 13, using DINOv2 as the teacher does not increase the firing rate, and our method achieves lower energy consumption on CIFAR100 than existing SNN distillation approaches.

### A.5 RECONSTRUCTION MODULE

To preserve low-level input information and mitigate the information loss in early spiking layers, we attach a lightweight **Reconstruction Module** to the stem output of the network. Given the time-

Table 8: Effect of auxiliary ANN adapters' convolutional layers (ImageNet-100, Backbone parameters 21.34M, $K$=128). Best results are in bold.

| ANN adapters' arch | #Conv Layers | Params (M) | Spike-ResNet-34 (%) |
|---|---|---|---|
| 7 | 1 | 24.33 | 69.16 |
| 3 | 1 | 22.69 | 69.46 |
| 7-3 | 2 | 24.77 | **69.82** |
| 7-5-3 | 3 | 26.00 | 69.60 |
| 7-5-3-3-3 | 4 | 26.44 | 66.96 |
| 7-7-5-5-5-3-3-3-3 | 8 | 31.75 | 55.46 |

Table 9: Effect of auxiliary ANN adapters' numbers on accuracy and parameters (ImageNet-100, $K = 128$). Best results are in **bold**.

| ANN Adapter Number | ResNet-34 | | | ResNet-101 | | |
|---|---|---|---|---|---|---|
| | Spike Acc (%) | MS Acc (%) | Params (M) | Spike Acc (%) | MS Acc (%) | Params (M) |
| 1 | 64.80 | 70.82 | 22.61 | 38.78 | 70.18 | 44.10 |
| 2 | 67.42 | 71.36 | 23.47 | 43.20 | 71.32 | 45.50 |
| 3 | **69.60** | **71.88** | 24.77 | **59.62** | **72.64** | 46.83 |

distributed feature

$$\mathbf{F}_0 \in \mathbb{R}^{T \times B \times C \times H' \times W'},$$

the module predicts a per-step reconstruction of the input:

$$\hat{\mathbf{X}} \in \mathbb{R}^{T \times B \times C_{\text{in}} \times H \times W}.$$

**Module Design.** The reconstruction head is a compact time-distributed decoder consisting of an upsampling layer and a projection layer. Its architecture is summarized in Table 14. The first layer upsamples the spatial resolution by a factor of 1 and reduces channels from $C$ to $C/2$, while the second layer maps features back to the RGB input space.

**Placement in the Network.** We attach the reconstruction head after the stem layer (conv0), where the feature still contains rich low-level information but has entered the spike-driven regime. It may also be optionally applied to deeper layers by specifying a reconstruction layer index.

**Computational Cost.** The reconstruction head introduces only two small convolutions. For typical settings ($C$=64, $C_{\text{in}}$=3, $H'$=$H/2$):

- Parameters: **0.04M–0.06M** (negligible compared to backbone).

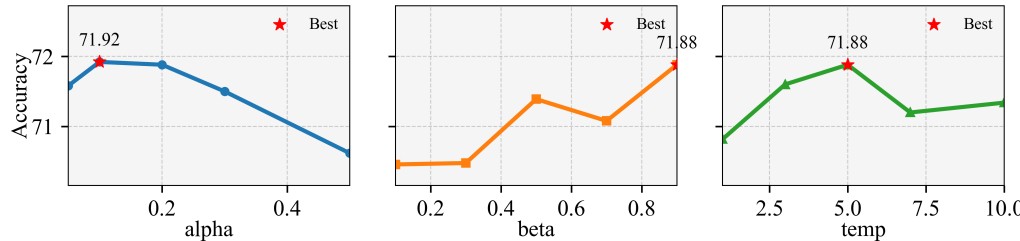

Figure 6: Sensitivity analysis of hyperparameters $\alpha$, $\beta$, and temperature on ImageNet100 with MS-ResNet-34.

Table 10: Effect of depth dependent auxiliary head weights on accuracy (ImageNet-100, Spike-ResNet-101). Best results are in bold.

| Setting | $w_c^{(1)}, w_k^{(1)}$ | $w_c^{(2)}, w_k^{(2)}$ | $w_c^{(3)}, w_k^{(3)}$ | Top-1 Acc. (%) |
|---------|------------------------|------------------------|------------------------|----------------|
| Baseline | 1.0 | 1.0 | 1.0 | 69.54 |
| | 0.8 | 1.0 | 1.2 | **70.04** |
| | 0.6 | 1.0 | 1.4 | 69.03 |
| | 0.4 | 1.0 | 1.6 | 68.64 |

Table 11: Effect of depth dependent temperature schedules on accuracy (ImageNet-100, Spike-ResNet-101). Best results are in bold.

| Setting | $\tau_1$ | $\tau_2$ | $\tau_3$ | Top-1 Acc. (%) |
|---------|----------|----------|----------|----------------|
| Baseline | 5 | 5 | 5 | 69.54 |
| Deeper heads sharper | 3 | 5 | 7 | 69.42 |
| Shallower heads sharper | 7 | 5 | 3 | **69.66** |

- FLOPs: $< 1\%$ of MS-ResNet-34/50/101 on ImageNet-1K.

- No additional time steps are introduced.

Thus, the module provides strong low-level information preservation at almost no computational overhead.

# B RESULTS ON CIFAR

## B.1 CIFAR-10/100

Since the effectiveness of full-spiking architectures has already been validated on ImageNet-1K, we focus on mixed-precision models (MS-ResNet-18/19) for experiments on CIFAR-10 and CIFAR-100, under time steps $T \in 1, 2, 4$. To ensure fair comparison, we adopt ResNet-18 and ResNet-19 as student networks, consistent with prior distillation studies. Unlike ImageNet-1K, DinoV2 does not provide a CIFAR-specific classification head, and direct inference requires resizing images from $32 \times 32$ to $224 \times 224$, which significantly increases computational overhead. Therefore, we employ ResNet-34 as the teacher model, trained following the same protocol as [refxx], achieving 97.14% top-1 accuracy.

The results in Table 15 show that our method achieves new state-of-the-art performance across all settings. Notably, the improvements at $T = 1$ are particularly significant, delivering competitive accuracy while reducing inference cost in terms of energy consumption, latency, and memory usage. These findings highlight the robustness and efficiency of our framework on smaller-scale datasets, further confirming its generalizability beyond large-scale ImageNet experiments.

## B.2 CIFAR-10DVS

We also evaluate our method on the neuromorphic CIFAR10-DVS dataset using the MS-ResNet-34 backbone. Our approach achieves the highest accuracy under both 4 and 10 timesteps, consistently outperforming prior SNN training methods across the board, as shown in Table **??**.

# C VISUALIZATION

Table 12: Effect of placing ANN adapters at different depths on accuracy (ImageNet-100, Spike-ResNet-101). Best results are in bold.

| Setting | position-1 | position-2 | position-3 | Top-1 Acc. (%) |
|---------|-----------|-----------|-----------|----------------|
| Baseline | 22 | 49 | 76 | 69.54 |
|  | 19 | 40 | 61 | **70.06** |
|  | 40 | 61 | 79 | 63.42 |
|  | 22 | 37 | 55 | 69.84 |
|  | 55 | 67 | 82 | 52.02 |

Table 13: Firing rate and energy comparison.

(a) Different Teachers on ImageNet1k with MS-ResNet-34

| Teacher | Firing rate (%) | Energy (mJ) |
|---------|-----------------|-------------|
| DINOv2-small | 20.40 | 0.765 |
| DINOv2-base | 20.52 | 0.769 |
| DINOv2-large | 20.56 | 0.771 |
| ResNet34 | 20.53 | 0.770 |

(b) Comparison with other methods on CIFAR100 with MS-ResNet-18

| Methods | Firing Rate(%) | Energy(mJ) |
|---------|----------------|------------|
| Vanilla SNN KD | 22.85 | 0.42 |
| BKDSNN | 34.93 | 0.52 |
| TSER | 22.95 | 0.42 |
| **Ours** | **21.25** | **0.39** |

## C.1 VISUALIZATION OF RECONSTRUCT IMAGE

As shown in Figure 7 , the layer-wise reconstructions show that direct training rapidly loses spatial and semantic fidelity as depth increases, whereas our method preserves recognizable structures even in the deepest layers of Spike-ResNet-101, demonstrating its ability to effectively mitigate information degradation and maintain higher-quality internal representations throughout the network.

## C.2 VISUALIZATION OF T-SNE

We conduct experiments on ImageNet100 and visualize the model outputs in Figure 8 using t-SNE (Van der Maaten & Hinton, 2008). We present the visualization in two scenarios, one obtained by direct training and the other by our method, with distinct colors representing different categories. As shown in Figure 8, the embeddings produced by our method (Figure 8(b)) exhibit tighter intra-class clustering and clearer inter-class separation compared to direct training (Figure 8(a)).

## C.3 VISUALIZATION OF SSIM

Following the analysis protocol of (Hu et al., 2024), we investigate the effect of our method on enhancing information flow in deep SNNs, specifically Spike-ResNet-101. SSIM defined as $SSIM(x,y) = \frac{(2\mu_x\mu_y+C_1)(2\sigma_{xy}+C_2)}{(\mu_x^2+\mu_y^2+C_1)(\sigma_x^2+\sigma_y^2+C_2)}$ , the similarity between adjacent blocks reflects the degree of redundancy in their representations: values approaching 1 indicate that consecutive layers learn nearly identical features, thereby failing to exploit the potential of deep architectures, whereas values moderately below 1 and distributed evenly suggest smooth and effective information propagation. As shown in Figure 9, the fully-spiking Spike-ResNet-101 suffers from severe information bottlenecks, with adjacent layers converging toward redundant representations. In contrast, our method substantially improves the diversity and smoothness of inter-block representations, which facilitates more effective information transmission and consequently leads to stronger performance in deep spiking networks.

## D PROOF OF PROPOSITION 1

Since here we consider the entropy capacity of a single layer, we omit the layer-index subscript $l$ in the sequel derivations for notational simplicity without any confusion. Considering spike-count coding, we denote $C = [c_1, \ldots, c_n]$ for $n$ neurons, where $c_i = \sum_{t=1}^{T} z_i^t \in \{0, \ldots, T\}$ is the spike

Table 14: Operations of the Reconstruction Module.

| Stage | Operation | Output Shape |
|---|---|---|
| Input | — | $(T, B, C, H', W')$ |
| Upsampling | Transposed Conv (stride 2) | $(T, B, C/2, H, W)$ |
| Normalization | Temporal BN + ReLU | $(T, B, C/2, H, W)$ |
| Projection | 3×3 Conv | $(T, B, C_{in}, H, W)$ |

Table 15: Comparison results on CIFAR-10/100, including direct training methods and knowledge distillation methods.

| Method | Model | TOP-1 ACC(%) | | | | | |
|---|---|---|---|---|---|---|---|
| | | CIFAR-10 | | | CIFAR-100 | | |
| **Direct training / Time-step** | | 1 | 2 | 4 | 1 | 2 | 4 |
| DspikeLi et al. (2021b) | ResNet-18 | - | 93.13 | 93.66 | - | 71.68 | 73.35 |
| STBP-tdBNZheng et al. (2021) | ResNet-19 | - | 93.13 | 93.66 | - | 71.68 | 73.35 |
| TETDeng et al. (2022) | ResNet-19 | - | 94.16 | 94.44 | - | 72.87 | 74.47 |
| RecDisGuo et al. (2022) | ResNet-19 | - | 93.64 | 95.53 | - | - | 74.10 |
| RateBPYu et al. (2024a) | ResNet-18 | - | 94.75 | 95.61 | - | 75.97 | 78.26 |
| | ResNet-19 | - | 96.23 | 96.26 | - | 79.87 | 80.71 |
| GLIFYao et al. (2022) | ResNet-18 | - | 94.15 | 94.67 | - | 74.60 | 76.42 |
| | ResNet-19 | - | 94.44 | 94.85 | - | 75.48 | 77.05 |
| **With Distillation / Time-step** | | 1 | 2 | 4 | 1 | 2 | 4 |
| KDSNN Xu et al. (2023) | ResNet-18 | - | - | 93.41 | - | - | - |
| Joint A-SNN Guo et al. (2023a) | ResNet-18 | - | 94.01 | 95.45 | - | 75.79 | 77.39 |
| | ResNet-34 | - | 95.13 | 96.07 | - | 77.11 | 79.76 |
| SM Deng et al. (2023) | ResNet-18 | - | - | 94.07 | - | 75.79 | 79.49 |
| | ResNet-19 | - | - | 96.82 | - | - | 81.70 |
| BKDSNN Xu et al. (2024) | ResNet-19 | - | - | 94.64 | - | - | 74.95 |
| TSSD Zuo et al. (2024) | ResNet-18 | - | 93.37 | - | - | 73.40 | - |
| TKS Dong et al. (2024) | ResNet-19 | - | - | 96.35 | - | - | 79.89 |
| EnOF Guo et al. (2024) | ResNet-19 | - | 96.19 | - | - | 82.43 | - |
| SuperSNN Zhang et al. (2024) | ResNet-19 | - | 95.08 | - | - | 76.49 | - |
| FRTD Yu et al. (2025b) | ResNet-18 | 94.27 | 95.11 | 95.57 | 74.34 | 77.32 | 79.10 |
| | ResNet-19 | 94.59 | 96.65 | 96.97 | - | 81.47 | 82.47 |
| TSER Yu et al. (2025c) | ResNet-18 | - | 95.58 | 96.18 | - | 78.30 | 79.69 |
| | ResNet-19 | - | 96.72 | - | - | 81.29 | - |
| **Our** | ResNet-18 | **94.85** | **95.73** | **96.48** | **75.24** | **78.81** | **79.59** |
| | ResNet-19 | **95.17** | **96.9** | **97.2** | **76.02** | **82.5** | **83.5** |

count of neuron $i$. We first upper-bound $H(C)$ by the sum of single-neuron entropy

$$H(C) = H(c_1, \ldots, c_n) \le \sum_{i=1}^{n} H(c_i), \tag{12}$$

where the equality holds if $c_i$'s are independent. Thus we have

$$\sup H(C) \le \sum_{i=1}^{n} \sup H(c_i). \tag{13}$$

Table 16: Performance comparison of top-1 accuracy (%) on CIFAR10-DVS.

| Method | Model | Timestep | Acc. (%) |
|--------|-------|----------|----------|
| STBP-tdBN  (Zheng et al., 2021) | ResNet-19 | 10 | 67.80 |
| Dspike  (Li et al., 2021a) | ResNet-18 | 10 | 75.40 |
| SM  (Deng et al., 2023) | ResNet-18 | 10 | 83.19 |
| FRTD  (Yu et al., 2025b) | ResNet-18 | 4 | 83.50 |
| FRTD  (Yu et al., 2025b) | ResNet-18 | 10 | 86.40 |
| **Ours** | ResNet-18 | 4 | **84.20** |
| **Ours** | ResNet-18 | 10 | **86.80** |

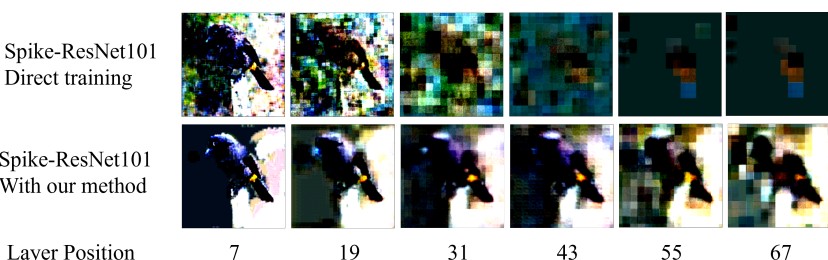

| | | | | | | |
|--|--|--|--|--|--|--|
| Spike-ResNet101 Direct training | | | | | | |
| Spike-ResNet101 With our method | | | | | | |
| Layer Position | 7 | 19 | 31 | 43 | 55 | 67 |

Figure 7: Comparison between direct training and our method for reconstructing the input images at different layers of Spike-ResNet-101.

For a neuron $i$, denote its firing probability at time-step $t$ by $p_{i,t} := P(z_i^t = 1)$ and its mean firing probability by $\bar{p}_i = \frac{1}{T}\sum_{t=1}^{T} p_{i,t}$. Since $c_i$ is a deterministic function of $z_i^{1:T}$, $c_i = \sum_{t=1}^{T} z_i^t$, we have

$$H(c_i) \leq H(z_1, \cdots, z_T) \leq \sum_{t=1}^{T} H(z_t) = -\sum_{t=1}^{T} p_{i,t}\log p_{i,t} + (1 - p_{i,t})\log(1 - p_{i,t}), \quad (14)$$

where the equality in the second inequality holds when $\{z_i^t\}$ are independent. Moreover, for a fixed mean firing probability $\bar{p}_i$, the entropy of $c_i$ is maximized when all the marginal distributions $p_{i,t}$'s are equal to $\bar{p}_i$, therefore we have

$$H(c_i) \leq H_{\mathrm{Bin}}(T, \bar{p}), \quad (15)$$

where the equality holds when $p_{1,t} = \cdots = p_{1,T} = \bar{p}_i$, and $H_{\mathrm{Bin}}$ is the entropy of Binomial distribution given by $H_{\mathrm{Bin}}(T, p) = -\sum_{k=0}^{T} \binom{T}{k} p^k (1-p)^{T-k} \log\left(\binom{T}{k} p^k (1-p)^{T-k}\right)$.

We further consider the firing-rate constraint

$$\frac{1}{nT}\sum_{i=1}^{n}\sum_{t=1}^{T} p_{i,t} \leq \rho, \quad (16)$$

which is equivalent to

$$\frac{1}{n}\sum_{i=1}^{n} \bar{p}_i \leq \rho, \quad (17)$$

where $\rho \in (0, 1)$. Since $H_{\mathrm{Bin}}(T, p)$ is a concave function of $p \in [0, 1]$ and is symmetric about $p = \frac{1}{2}$, the sum $\sum_{i=1}^{n} H_{\mathrm{Bin}}(T, \bar{p}_i)$ under the constraint (17) is maximized when all the neurons have the same mean firing probability of $\min\{\rho, \frac{1}{2}\}$, i.e., $\bar{p}_1 = \cdots = \bar{p}_n = \min\{\rho, \frac{1}{2}\}$. Thus, denote $\tilde{\rho} = \min\{\rho, \frac{1}{2}\}$, we have

$$\sum_{i=1}^{n} \sup H(c_i) \leq \sum_{i=1}^{n} H_{\mathrm{Bin}}(T, \tilde{\rho}) = n H_{\mathrm{Bin}}(T, \tilde{\rho}), \quad (18)$$

which together with (13) and the fact that $H(c_i) \leq \log(T+1)$ results in (4) in Proposition 1.

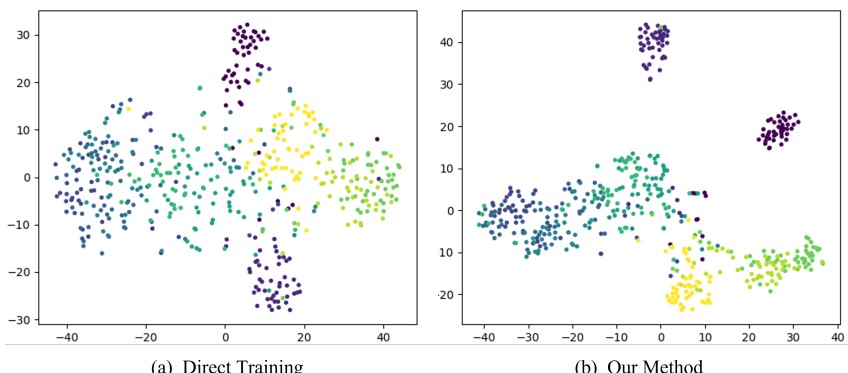

(a) Direct Training  (b) Our Method

Figure 8: T-SNE visualization of the direct training method and our method with Spike-ResNet-34 on ImageNet100.

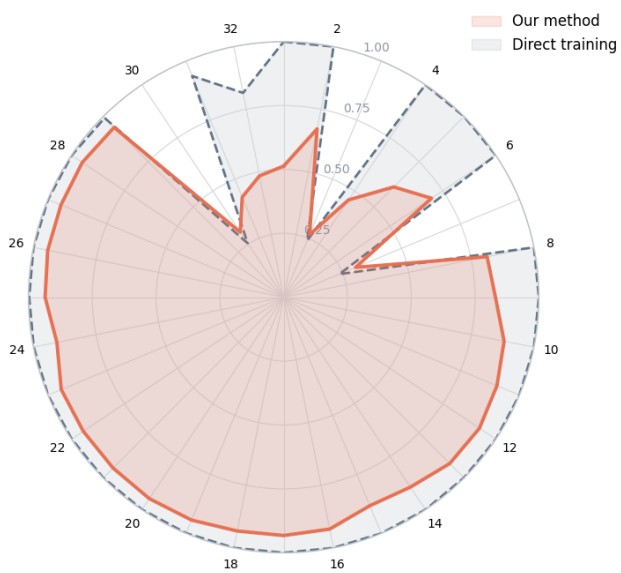

Figure 9: Radar charts of SSIM for Spike-ResNet-101 trained by the direct training method and our method.

In the asymptotic case of $T \to \infty$, the entropy of Binomial distribution can be expressed as (Jacquet & Szpankowski, 1999)

$$H_{\mathrm{Bin}}(T, p) = \frac{1}{2} \log(2\pi e\, T\, p(1-p)) + O(1/T), \tag{19}$$

which together with $p = \tilde{\rho}$ and considering $n_l$ neurons lead to the asymptotic entropy capacity (5) in Proposition 1.

## E  PROOF OF PROPOSITION 2

From the data process chain $X \to Z_1^{1:T} \to \cdots \to Z_L^{1:T} \to C_L$, we have

$$I(X; C_L) \leq I(X; Z_L^{1:T}) \leq H(Z_L^{1:T}), \tag{20}$$

and

$$I(X; C_L) \leq H(C_L). \tag{21}$$

For layer $L$ with $n_L$ neurons and a firing rate constraint $\rho_L$, denote $\tilde{\rho}_L = \min\{\rho, \frac{1}{2}\}$, then from Proposition 1 we have

$$H(C_L) \leq n_L H_{\mathrm{Bin}}(T, \tilde{\rho}_L) \leq n_L \log(T+1), \tag{22}$$

which together with (21) results in

$$I(X; C_L) \leq H(C_L) \leq n_L H_{\mathrm{Bin}}(T, \tilde{\rho}_L) \leq n_L \log(T+1). \tag{23}$$

Then the asymptotic expression (6) in Proposition 2 is derived by using the asymptotic entropy of Binomial distribution $H_{\mathrm{Bin}}(T, p) = \frac{1}{2} \log(2\pi e T p(1-p)) + O(\frac{1}{T})$.

Further, under the assumption of the count-sufficiency condition of the data process chain $X \rightarrow C_1 \rightarrow \cdots \rightarrow C_L$, i.e., $I(Z_l^{1:T}; C_L | C_l) = 0$, it follows that for any $l \leq L$

$$I(X; C_L) \leq I(X; C_l), \tag{24}$$

which together with Proposition 1 results in

$$I(X; C_L) \leq I(X; C_l) \leq H(C_l) \leq n_l H_{\mathrm{Bin}}(T, \tilde{\rho}_l). \tag{25}$$

Finally, the inequality (7) in Proposition 2 can be derived.

