# OpenReview forum: "Breaking Information Impedance in Deep Spiking Neural Networks via Multi-Stage Foundation-Model Distillation"
_ICLR.cc/2026/Conference — Submitted to ICLR 2026_

### Official Review · Reviewer_DrHd · 2025-10-28

**Soundness:** 3
**Presentation:** 3
**Contribution:** 3
**Rating:** 6
**Confidence:** 4

**Summary:**

This paper attributes the difficulty of training deep SNNs to "information impedance" and proposes a multi-stage distillation framework to solve it. Using a foundation model DINOv2 to inject knowledge at multiple network stages, the method enhances information flow and achieves a new state-of-the-art accuracy on ImageNet with ResNet architectures, marking an advance for the field.

**Strengths:**

1. Principled and insightful problem analysis. The paper's primary strength is its rigorous, information-theoretic framing of the core difficulty in training deep SNNs. The concept of "information impedance" is intuitive and powerfully supported by both theoretical propositions and compelling empirical evidence. This analysis moves the field beyond heuristic solutions and provides a clear, fundamental reason for why deep SNNs are hard to train, which is a significant contribution in itself.
2. Well-motivated and effective method: The proposed multi-stage distillation framework is a direct and logical solution to the identified problem of information impedance and its effectiveness is demonstrated through experiments.
3. Well-written, clearly structured, and easy to follow.

**Weaknesses:**

1. About novelty of methodological components: While the application and justification are highly novel, the core components of the method—multi-stage supervision (i.e., deep supervision [1,2]) and knowledge distillation with auxiliary branches [3,4] —are established techniques in the broader deep learning literature. The paper would be slightly strengthened by explicitly acknowledging this and framing its contribution as the novel synthesis and principled application of these techniques to solve a specific, fundamental problem in SNNs.
2. Contextualization of the Information-Theoretic Perspective: While the analysis from an information-theoretic viewpoint is a key strength, the paper could better contextualize this contribution by expanding its review of related literature. The concept of "information impedance" is closely related to the Information Bottleneck principle, which has been used to analyze deep ANNs [5,6,7]. Discussing more of this prior work would help situate the paper's theoretical framing within a broader context and more clearly highlight its specific insights for SNNs.




[1] Deeply-supervised nets. Artificial intelligence and statistics. PMLR, 2015.

[2] Be your own teacher: Improve the performance of convolutional neural networks via self distillation. ICCV, 2019.

[3] Distilling knowledge via knowledge review. CVPR, 2021.

[4] Decoupling dark knowledge via block-wise logit distillation for feature-level alignment. TAI, 2024.

[5] On the information bottleneck theory of deep learning, 2019.

[6] Revisiting Locally Supervised Learning: an Alternative to End-to-end Training. ICLR, 2021.

[7] Go beyond End-to-End Training: Boosting Greedy Local Learning with Context Supply. 2023.

**Questions:**

1. About the training overhead and trade-offs. The proposed method introduces auxiliary ANN adapters and reconstruction modules, which presumably add to the computational overhead during training. I am curious about the increase in training time (e.g., wall-clock time per epoch) and memory consumption compared to the baseline direct training and vanilla KD setups. A discussion on the trade-off between this increased training cost and the significant performance gains would be valuable, as it would help researchers assess the practical costs of implementing the approach.

2. On the choice of teacher model. The choice of DINOv2 as a powerful, pre-trained foundation model is a key aspect of your framework. Many prior SNN-KD works, however, use an ANN counterpart with an identical architecture as the teacher (e.g., an ANN ResNet-34 teaching a spiking ResNet-34). Could you comment on the decision to exclusively use a foundation model? How do you expect the results might differ if a standard, trained-from-scratch ANN counterpart were used as the teacher? This would help to disentangle the benefits of the multi-stage distillation architecture itself from the benefits derived from the rich, pre-existing knowledge of the foundation model.

---

> ### Author Response · Authors · 2025-11-25
>
> We sincerely thank the reviewer for the valuable and constructive comments, here we provide clarifications and additional results to address your concerns.
> ### 1.  About novelty of methodological components: While the application and justification are highly novel, the core components of the method—multi-stage supervision (i.e., deep supervision [1,2]) and knowledge distillation with auxiliary branches [3,4]—are established techniques in the broader deep learning literature. The paper would be slightly strengthened by explicitly acknowledging this and framing its contribution as the novel synthesis and principled application of these techniques to solve a specific, fundamental problem in SNNs.
> Thanks very much for this constructive suggestion and pointing out these related works [Ref1-Ref4]. In the revised manuscript, we have added a discussion to explicitly acknowledge these related works in Section 2 (Related Work).
>
> **While our multi-stage KD resemble deep supervision [Ref1, Ref2] and KD with auxiliary branches [Ref3, Ref4] in form, our contribution lies not only in architectural design but in a theory-driven motivation and mechanism specifically tailored for deep SNNs, whose effectiveness is strongly and consistently validated by our extensive empirical results on deep SNNs.**
>
> **(1) Comparison with deep supervision and KD with auxiliary branches**. Deep supervision [Ref1] typically adds intermediate CE with one-hot labels mainly for optimization/gradient shortcutting. Self-distillation [Ref2] divides a network into several stages and distills knowledge within network itself, by which the knowledge in the deeper portion of the networks is squeezed into the shallow ones. KD with auxiliary branches [Ref3-4] also consider stage separation and use auxiliary branches for cross-stage distillation or better feature alignment. In contrast, **our intermediate heads distill high-entropy teacher posteriors from a foundation model via local KL, particularly targeting information impedance of deep SNNs.** Our experiments show that (see Table I, also Table 3 in the manuscript):
> - **For ANN, vanilla KD is sufficient** and multi-stage KD does not yield any improvement, which is consistent with previous results [Ref5].
> - **For both fully-spiking and hybrid SNNs, our method achieves significant improvement over the vanilla method.** The improvement is particularly pronounced for the full-spiking ResNet.
>
> The proposed method demonstrates particular effectiveness on deeper SNN architectures. For Spike-ResNet-101, which suffers from pronounced information degradation under its fully-spiking design, our method achieves a remarkable improvement of +59.58\% over direct training. Overall, these results clearly demonstrate the effectiveness of our method in alleviating the information transmission bottleneck in deep SNNs.
>
> #### Table I. Comparison of accuracy (%) on ImageNet100 across different architectures and depths.
> | Method                 | Spike-ResNet-34      | ResNet34 (ANN)      | Spike-ResNet-50      | Spike-ResNet-101      |
> |------------------------|------------------------|---------------------------|------------------------|-------------------------|
> | Baseline| 71.50  | 82.58  | 69.85  | 9.78  |
> | $𝐿_{kd}$| 77.04 (+5.54)    | **87.76 (+5.18)**       | 77.36 (+7.51)     | 15.60 (+5.82)          |
> | w/o $L_{rec}$| 80.32 (+8.82)   | 87.65 (+5.02) | 80.26 (+10.41)   | 68.76 (+58.98)         |
> | **Ours**| **80.81 (+9.31)**     | 87.53 (+4.95)  | **80.86 (+11.01)**    | **69.36 (+59.58)**      |
>
> **(2) Comparison with feature-alignment KD**. Most prior SNN KD methods follow ANN-style logit matching or intermediate feature alignment [Ref6, Ref7]. **For SNNs, the performance of intermediate feature alignment based KD is limited by severe ANN-SNN feature heterogeneity** (continuous vs. spiking representations). **Our lightweight ANN adapters map spiking features to distillable logits, and we align teacher posteriors without hard intermediate feature matching, which is explicitly tailored to overcome impedance of SNNs.**
>
> - *[Ref1] C. Lee, et al. Deeply supervised nets. AISTATS, 2015.*
> - *[Ref2] L. Zhang. Be your own teacher: Improve the performance of convolutional neural networks via self distillation. ICCV 2019.*
> - *[Ref3] P. Chen. Distilling knowledge via knowledge review. CVPR  2021.*
> - *[Ref4] C. Yu. Decoupling dark knowledge via block-wise logit distillation for feature-level alignment. IEEE TAI, 2024b.*
> - *[Ref5] Z. Hao, et al. Revisit the power of vanilla knowledge distillation: from small scale to large scale. NeurIPS, 2023.*
> - *[Ref6] BKDSNN: Enhancing the Performance of Learning-based Spiking Neural Networks Training with Blurred Knowledge Distillation, ECCV, 2024.*
> - *[Ref7] D. Hong, et al. Lasnn: Layer-wise ann-to-snn distillation for effective and efficient training in deep spiking neural networks. arXiv preprint ,2023.*

---

> ### Author Response · Authors · 2025-11-25
>
> ### 2. Contextualization of the Information-Theoretic Perspective: While the analysis from an information-theoretic viewpoint is a key strength, the paper could better contextualize this contribution by expanding its review of related literature. The concept of “information impedance” is closely related to the Information Bottleneck principle, which has been used to analyze deep ANNs [5,6,7]. Discussing more of this prior work would help situate the paper’s theoretical framing within a broader context and more clearly highlight its specific insights for SNNs.
> Thank you for pointing out these related works [5-7]. In the revised manuscript, we have added a discussion in Section 4.2 to better contextualize our information-theoretic perspective within the broader literature.
>
> The works [6] and [7] apply the Information Bottleneck (IB) principle [5] to analyze deep ANNs in the context of locally supervised learning, focusing on how local objectives shape representation compression and task-relevant information. In contrast, **our analysis targets the information flow in SNNs, emphasizing the unique information impedance induced by spike-based representations. Specifically, we characterize how quantization-to-spikes and the cumulative depth of fully spiking architectures jointly lead to progressive task-information attenuation, a phenomenon that differs structurally from ANN compression behavior observed in IB analyses.** Our multi-stage distillation is then derived to explicitly mitigate this spike-induced impedance rather than to provide local supervised training per se.
>
> ### 3. About the training overhead and trade-offs. The proposed method introduces auxiliary ANN adapters and reconstruction modules, which presumably add to the computational overhead during training. I am curious about the increase in training time (e.g., wall-clock time per epoch) and memory consumption compared to the baseline direct training and vanilla KD setups. A discussion on the trade-off between this increased training cost and the significant performance gains would be valuable, as it would help researchers assess the practical costs of implementing the approach.
> Thank you for your suggestion. We have compared the training overhead of direct training, vanilla KD (ANN ResNet teacher of the same architecture), vanilla KD (DINOv2-base), and our method for different SNN ResNet students, as shown in the following Table II. Compared with vanilla KD (DinoV2), **our method with auxiliary ANN adapters introduces only a marginal increase in training time and memory usage**. Using DINOv2-base as the teacher introduces an increase in training time. However, the relative overhead decreases as the network depth grows. We have added these results into Table 4 in the manuscript.
> #### Table II. Training cost comparison under different teacher models on ImageNet1k (batchsize=1024, DINOv2-base denotes the DINOv2-base teacher, ResNet denotes the ANN teacher of the same depth).
> | Methods             | MS-ResNet34 (Time)      | MS-ResNet34 (Mem, GB)  | MS-ResNet50 (Time)      | MS-ResNet50 (Mem, GB)  | MS-ResNet101 (Time)     | MS-ResNet101 (Mem, GB) |
> |---------------------|--------------------|---------------------|---------------------|---------------------|----------------------|----------------------|
> | Direct Train        | 6m52s              | 78.7                | 10m45s              | 167.5               | 14m53s               | 228.1                |
> | Vanilla KD (ResNet) | 7m38s (+11.2%)     | 80.5 (+2.3%)        | 11m29s (+6.8%)      | 169.9 (+1.4%)       | 16m34s (+11.3%)      | 230.3 (+1.0%)        |
> | Vanilla KD (DINOv2) | 9m29s (+38.1%)     | 89.8 (+14.1%)       | 13m41s (+27.2%)     | 173.4 (+3.5%)       | 18m22s (+23.4%)      | 235.4 (+3.2%)        |
> | Our Method (DINOv2) | 10m08s (+47.6%)    | 91.0 (+15.6%)       | 14m23s (+33.7%)     | 176.4 (+5.3%)       | 19m10s (+28.8%)      | 240.4 (+5.4%)        |

---

> ### Author Response · Authors · 2025-11-25
>
> ### 4. On the choice of teacher model. The choice of DINOv2 as a powerful, pre-trained foundation model is a key aspect of your framework. Many prior SNN-KD works, however, use an ANN counterpart with an identical architecture as the teacher (e.g., an ANN ResNet-34 teaching a spiking ResNet-34). Could you comment on the decision to exclusively use a foundation model? How do you expect the results might differ if a standard, trained-from-scratch ANN counterpart were used as the teacher? This would help to disentangle the benefits of the multi-stage distillation architecture itself from the benefits derived from the rich, pre-existing knowledge of the foundation model.
>
> Thanks for this insightful comment. We have conducted additional evaluation to compare with the KD method using an ANN counterpart with an identical architecture as the teacher.
>
> As shown in Table III and IV, for a MS-ResNet-34 student, using a large DinoV2 as the teacher yields better performance than ResNet34 on ImageNet-1k, meanwhile it delivers substantially stronger performance on robustness benchmarks such as ImageNet-C and fine-grained recognition tasks, which addresses a gap in prior work that ignored robustness. We have updated these results in Table 2 in the revised manuscript.
>
> #### Table III. Performance (MS-ResNet-34, T=1, 50-epoch training) on ImageNet-1k and ImageNet-C under different teachers.
> | Teacher Model | Parameters | ImageNet-1k| Tested on ImageNet-C (severity-5) |
> |---------------|------------|----------|------------|
> | DinoV2-small      | 21M        | 65.40%   | 22.94%     |
> | DinoV2-base     | 86M        | 65.95%   | 23.40%     |
> | DinoV2-large      | 300M       | 65.89%   | 23.93%     |
> | ResNet34        | 21.8M       | 65.33%   | 21.66%     |
>
> #### Table IV. Performance (MS-ResNet-34, T=1, 50-epoch training) on fine-grained datasets under different teachers.
> | Teacher Model | Parameters |  Cars   | Birds  | Dogs   | Aircraft | Avg.   |
> |---------------|------------|--------|--------|--------|----------|--------|
> | DinoV2-small      | 21M        | 42.13% | 44.60% | 70.49% | 42.69%   | 49.98% |
> | DinoV2-base     | 86M        | 42.57% | 45.03% | 70.22% | 42.88%   | 50.18% |
> | DinoV2-large      | 300M       |  43.12% | 45.49% | 70.92% | 43.58%   | 50.78% |
> | ResNet34        | 21.8M       |  41.71% | 42.61% | 68.76% | 41.21%   | 48.57% |

---

### Official Review · Reviewer_L8b7 · 2025-10-30

**Soundness:** 1
**Presentation:** 2
**Contribution:** 1
**Rating:** 2
**Confidence:** 5

**Summary:**

This paper proposes a novel multi-stage knowledge distillation framework (MSKD-SNN) to mitigate the information impedance that limits the training of deep spiking neural networks (SNNs). The authors present a theoretical analysis based on information theory, derive entropy bottlenecks across layers, and validate their claim empirically. They introduce a multi-stage distillation pipeline leveraging a foundation model teacher (DINOv2) and auxiliary ANN adapters, showing significant gains on ImageNet-1K (77.14% Top-1 with MS-ResNet-101).

**Strengths:**

The authors clearly identify information impedance as a bottleneck in deep SNNs, where representational capacity diminishes due to spiking quantization and sparse activations. It provides an insightful information-theoretic formulation and two propositions quantifying representational entropy capacity and layer-wise bottlenecks. The theoretical framing is well articulated and provides a fresh analytical lens for SNN limitations.

**Weaknesses:**

1. While the paper introduces “information impedance” as a concept, it largely reformulates known ideas about information bottlenecks in quantized or discrete neural systems. The propositions (Sec. 4.1–4.2) rely on standard mutual information reasoning without new derivations beyond entropy bounds.
2. The proposed multi-stage KD resembles hierarchical distillation or intermediate feature matching, previously explored in ANN and hybrid SNN settings (e.g., TKS, EnOF, FRTD). The distinction mainly lies in terminology (“information impedance”) rather than methodological novelty.
3. The use of DINOv2 as teacher makes the approach less generalizable to domains where large pretrained teachers are unavailable. This dependency somewhat contradicts the neuromorphic efficiency motivation, since DINOv2-based supervision is computationally heavy.
4. Although the paper claims to improve “training efficiency,” no quantitative measurement of energy, latency, or FLOPs is provided. The auxiliary ANN adapters and multiple distillation heads add nontrivial computation overhead, which is not analyzed.
5. The empirical evaluation uses a reconstruction proxy, but details such as reconstruction network architecture, regularization, or sampling variance are omitted. The reliability of the “information flow” estimation remains uncertain.
6. Results are restricted to conventional GPUs. No neuromorphic hardware tests (e.g., Loihi, Tianjic) are conducted, despite claims about energy-efficient learning. It remains unclear whether the auxiliary ANN modules are implementable in spike-based hardware.
7. The term “breaking information impedance” suggests a paradigm shift, but the actual improvement is incremental. The language throughout could be moderated to match the scale of contribution.

**Questions:**

1. How sensitive is the proposed framework to the number and depth of auxiliary adapters?
2. Can the distillation heads be pruned after training to reduce inference cost?
3. Does the reconstruction module affect spike sparsity or latency during inference?
4. How would the method perform if the teacher is another SNN rather than an ANN?
5. Can “information impedance” be quantitatively estimated during training to adaptively guide loss weights?

---

> ### Author Response · Authors · 2025-11-25
>
> We sincerely thank the reviewer for the detailed and valuable comments, below we provide explanations and additional experiments to address your concerns.
> ### 1. While the paper introduces “information impedance” as a concept, it largely reformulates known ideas about information bottlenecks in quantized or discrete neural systems. The propositions (Sec. 4.1–4.2) rely on standard mutual information reasoning without new derivations beyond entropy bounds.
> Thanks very much for the comment. We agree that MI and IB perspectives have been discussed for quantized or discrete networks in general, but our contribution is not a mere rephrasing. We provide an SNN-specific impedance characterization for deep spike-count models and validate it systematically.
> - (1) Under spike-count coding with firing-rate constraints, we derive a closed-form entropy-capacity bound for a spiking layer (Proposition 1), which shows only logarithmic scaling with time-steps $T$, a result tied to SNN counting statistics rather than generic quantization.
> - (2) Proposition 2 shows the end-to-end MI is clamped by the minimum layer capacity, and that depth introduces additional bottlenecks because **firing-rate attenuation further tightens the bound. This leads to cumulative “information impedance” with depth, which explains the degradation of deep fully-spiking SNNs.**
> - (3) We then empirically estimate $I(X;Z_l)$ across deep Spike/MS/SEW-ResNets (34/50/101) and confirm sharper decay for deeper and fully-spiking models, especially at small $T$, which matches the analysis.
>
> These analyses directly motivate our multi-stage distillation that **decomposes a high-impedance chain into low-impedance stages and injects high-entropy teacher posteriors at checkpoints**.

---

> ### Author Response · Authors · 2025-11-25
>
> ### 2. The proposed multi-stage KD resembles hierarchical distillation or intermediate feature matching, previously explored in ANN and hybrid SNN settings (e.g., TKS, EnOF, FRTD). The distinction mainly lies in terminology (“information impedance”) rather than methodological novelty.
> Thanks for the comment. The methods TKS and FRTD mainly consider temporal knowledge distillation, whilst EnOF considers using a well-trained ANN to guide and improve the learning of the output feature representation of an SNN. **While our multi-stage KD resemble some existing hierarchical distillation in form, our contribution lies not only in architectural design but in a theory-driven motivation and mechanism specifically tailored for deep SNNs, whose effectiveness is strongly and consistently validated by our extensive empirical results on deep SNNs.**
>
> In the revised manuscript, we have added a detailed discussion in Section 2 (Related Work) to clarify the difference from existing multi-stage distillation methods.
>
> **(1) Comparison with existing hierarchical distillation.** Our multi-stage KD method resembles deep supervision [Ref1-2] and KD with auxiliary branches  [Ref3]  in form. Deep supervision  [Ref1]  adds intermediate CE loss with one-hot labels mainly for optimization shortcutting. Self-distillation  [Ref2]  divides a network into several stages and distills knowledge within network itself.  [Ref3]  also consider stage separation and use auxiliary branches for cross-stage distillation and better feature alignment. In contrast, our intermediate heads distill high-entropy teacher posteriors from a foundation model, **particularly targeting information impedance of deep SNNs**. Importantly, our results show that (Table I):
> - **For ANN, vanilla KD is sufficient and multi-stage KD does not yield any improvement**, which is consistent with previous results  [Ref4].
> - **For both fully-spiking and hybrid SNNs, our method achieves significant improvement over vanilla KD, particularly for deeper and full-spiking models.**
>
> Our method demonstrates particular effectiveness on deeper SNN architectures. As shown in Table I, for Spike-ResNet-101, which suffers from pronounced information degradation under its fully-spiking design, our method achieves a remarkable improvement of +59.58\% over direct training. Overall, these results clearly demonstrate the effectiveness of our method in alleviating the information transmission bottleneck in deep SNNs.
> #### Table I. Comparison of accuracy (%) on ImageNet100 across different architectures and depths.
> | Method                 | Spike-ResNet-34      | ResNet34 (ANN)      | Spike-ResNet-50      | Spike-ResNet-101      |
> |------------------------|------------------------|---------------------------|------------------------|-------------------------|
> | Baseline               | 71.50                 | 82.58               | 69.85                 | 9.78                   |
> | $𝐿_{kd}$                   | 77.04 (+5.54)         | **87.76 (+5.18)**       | 77.36 (+7.51)         | 15.60 (+5.82)          |
> | w/o $L_{rec}$              | 80.32 (+8.82)         | 87.65 (+5.02)       | 80.26 (+10.41)        | 68.76 (+58.98)         |
> | **Ours**               | **80.81 (+9.31)**     | 87.53 (+4.95)        | **80.86 (+11.01)**    | **69.36 (+59.58)**      |
>
> **(2) Different from intermediate feature matching based KD**. Most prior SNN KD methods follow ANN-style logit matching or intermediate feature alignment [Ref5, Ref6]. **For SNNs, the performance of intermediate feature matching based KD is limited by severe ANN-SNN feature heterogeneity** (continuous vs. spiking representations). **Our lightweight ANN adapters map spiking features to distillable logits, and we align teacher posteriors without hard intermediate feature matching, which is explicitly tailored to overcome impedance.**
>
> Empirically, as shown in Table 1 in the manuscript, our method significantly outperforms SOTA SNN KD baselines, and the gain is particularly significant on deep fully-spiking models, which matches our analysis. Particularly, on ImageNet-1k, **our method outperforms existing SOTA results by more than 4% on spike-ResNet-101 (72.86\% vs. 68.38\%).**
>
> - *[Ref1] C. Lee, et al. Deeply supervised nets. AISTATS, 2015.*
> - *[Ref2] L. Zhang. Be your own teacher: Improve the performance of convolutional neural networks via self distillation. ICCV 2019.*
> - *[Ref3] P. Chen. Distilling knowledge via knowledge review. CVPR  2021.*
> - *[Ref4] Z. Hao, et al. Revisit the power of vanilla knowledge distillation: from small scale to large scale. NeurIPS, 2023.*
> - *[Ref5] D. Hong, et al. Lasnn: Layer-wise ann-to-snn distillation for effective and efficient training in deep spiking neural networks. arXiv preprint ,2023.*
> - *[Ref6] BKDSNN: Enhancing the Performance of Learning-based Spiking Neural Networks Training with Blurred Knowledge Distillation, ECCV, 2024*

---

> ### Author Response · Authors · 2025-11-25
>
> ### 3. The use of DINOv2 as teacher makes the approach less generalizable to domains where large pretrained teachers are unavailable. This dependency somewhat contradicts the neuromorphic efficiency motivation, since DINOv2-based supervision is computationally heavy. Although the paper claims to improve “training efficiency,” no quantitative measurement of energy, latency, or FLOPs is provided. The auxiliary ANN adapters and multiple distillation heads add nontrivial computation overhead, which is not analyzed.
> Thanks very much for the comment. While using DINOv2 as a teacher increases training cost, this overhead occurs only during training and does not increase the energy consumption of SNN inference. Here we provide analysis on the **training cost** and **inference energy**.
>
> **(1) Training cost comparison.**
> The following Table II compares the training cost under different teacher models on ImageNet1k, including DINOv2-base teacher and ANN ResNet teachers of the same depth as the students. We compared the training overhead of direct training, vanilla KD (ANN ResNet teacher of the same architecture), vanilla KD (DINOv2-base), and our method for different SNN ResNet students. It can be seen that **Using DINOv2-base as the teacher introduces an increase in training time. However, the relative overhead decreases as the network depth grows. Compared with vanilla KD (DinoV2), our method using auxiliary ANN adapters introduces only a marginal increase in training time and memory usage.**  We have added this comparison into Table 4 in the revised manuscript.
> #### Table II. Training cost comparison under different teacher models on ImageNet1k (batchsize=1024, DINOv2-base denotes the DINOv2-base teacher, ResNet denotes the ANN teacher of the same depth).
> | Methods             | ResNet34 Time      | ResNet34 Mem (GB)  | ResNet50 Time      | ResNet50 Mem (GB)  | ResNet101 Time     | ResNet101 Mem (GB) |
> |---------------------|--------------------|---------------------|---------------------|---------------------|----------------------|----------------------|
> | Direct Train        | 6m52s              | 78.7                | 10m45s              | 167.5               | 14m53s               | 228.1                |
> | Vanilla KD (ResNet) | 7m38s (+11.2%)     | 80.5 (+2.3%)        | 11m29s (+6.8%)      | 169.9 (+1.4%)       | 16m34s (+11.3%)      | 230.3 (+1.0%)        |
> | Vanilla KD (DINOv2) | 9m29s (+38.1%)     | 89.8 (+14.1%)       | 13m41s (+27.2%)     | 173.4 (+3.5%)       | 18m22s (+23.4%)      | 235.4 (+3.2%)        |
> | Our Method (DINOv2) | 10m08s (+47.6%)    | 91.0 (+15.6%)       | 14m23s (+33.7%)     | 176.4 (+5.3%)       | 19m10s (+28.8%)      | 240.4 (+5.4%)        |
>
> **(2) Inference energy comparison.**
> We have conducted an analysis on the inference energy consumption. The following Table III compares the energy consumption of MS-ResNet-34 trained with different teachers on ImageNet-1K, whilst Table IV compares the energy consumption with other methods on CIFAR-100 using MS-ResNet-18. As shown in Table III and IV, **our method using DINOv2 as the teacher does not increase the firing rate and energy consumption of the model during inference.** Our method can achieves energy savings comparable to direct training and standard KD.
> #### Table III.  Energy comparison of MS-ResNet-34 trained with different teachers on ImageNet-1K.
> | Teacher model | Firing rate (%) | Energy (mJ / img) |
> |--------------|-----------------|--------------------|
> | DinoV2-small     | 20.40           | 0.765              |
> | DinoV2-base     | 20.52           | 0.769              |
> | DinoV2-large    | 20.56           | 0.771              |
> | ResNet34     | 20.53           | 0.770              |
>
> #### Table IV.  Energy comparison with other methods on CIFAR-100 using MS-ResNet-18.
> | Methods        | Firing Rate | Energy |
> |----------------|-------------|--------|
> | Vanilla SNN KD | 22.85%      | 0.42mJ |
> | BKDSNN         | 34.93%      | 0.52mJ |
> | TSER           | 22.95%      | 0.42mJ |
> | **Ours**       | 21.25%      | 0.39mJ |

---

> ### Author Response · Authors · 2025-11-25
>
> ### 4. The empirical evaluation uses a reconstruction proxy, but details such as reconstruction network architecture, regularization, or sampling variance are omitted. The reliability of the “information flow” estimation remains uncertain.
> Thank you for the insightful comment. The reconstruction network architecture is specified in Table V (also added in Appendix of the manuscript). We use the same reconstruction module and training setup for all probed models, so it serves as a unified probe that differences in reconstruction quality primarily reflect differences in the underlying representations rather than decoder capacity.
>
> We agree that reconstruction-based estimates of “information flow” are approximate and subject to variance, so we explicitly treat them as a relative proxy rather than an exact mutual-information value. In our experiments, the estimated trends are very consistent with architectural factors (such as depth, fully-spiking vs. hybrid) and with downstream performance. It indicates that this proxy is reliable enough for relatively comparing information flow across models and depths.
>
> Moreover, we have presented visualization of reconstructed images in C.1 of Appendix (Figure 7). The result shows that direct training rapidly loses spatial and semantic fidelity as depth increases, whereas our method preserves recognizable structures even in the deepest layers of Spike-ResNet-101, which demonstrates its ability to effectively mitigate information degradation and maintain higher-quality internal representations throughout the network.
> #### Table V.  Reconstruction network architecture.
> | Stage         | Operation                     | Output Shape              |
> |---------------|-------------------------------|----------------------------|
> | Input         | —                             | (T, B, C, H′, W′)          |
> | Upsampling    | Transposed Conv (stride 2)    | (T, B, C/2, H, W)          |
> | Normalization | Temporal BN + ReLU            | (T, B, C/2, H, W)          |
> | Projection    | 3×3 Conv                      | (T, B, C_in, H, W)         |
>
>
> ### 5. Results are restricted to conventional GPUs. No neuromorphic hardware tests (e.g., Loihi, Tianjic) are conducted, despite claims about energy-efficient learning. It remains unclear whether the auxiliary ANN modules are implementable in spike-based hardware.
> We are sorry for the confusion, **the auxiliary ANN modules are only used during training and are completely removed during inference.** Our inference pipeline is identical to other SNN distillation methods, incurring no additional computation overhead. At inference time, the model operates purely as an SNN.
> ### 6. The term “breaking information impedance” suggests a paradigm shift, but the actual improvement is incremental. The language throughout could be moderated to match the scale of contribution.
> Thanks for this suggestion. We have revised the title from “Breaking Information Impedance” to **“Mitigating Information Impedance”**, which more accurately reflects the scale of our contribution.

---

> ### Author Response · Authors · 2025-11-25
>
> ### 7. How sensitive is the proposed framework to the number and depth of auxiliary adapters?
> Thank you for the comment. The ablation study on the depth and number of auxiliary adapters is provided in Table 8 and Table 9 in the Appendix in our manuscript, which we also provide here in the following Table VI and Table VII.
>
> **Effect of the depth of auxiliary adapters.** The results show that configurations with only one or two convolutional layers are sufficient for achieving satisfactory performance, whereas deeper adapters ($\geq 4$ layers) lead to a clear performance drop. Two layers provide the best trade-off between accuracy gains and computational cost.
>
> **Effect of the number of auxiliary adapters.** For shallower networks (e.g., ResNet-34), adding ANN adapters initially improves accuracy, but the performance gain saturates beyond three adapters. In contrast, deep networks (e.g., ResNet-101) consistently benefit from additional ANN adapters, with higher relative improvement than shallow counterparts.
> #### Table VI. Effect of auxiliary ANN adapters’ convolutional layers (ImageNet-100, spike-ResNet-34 (21.34M), K=128). Best results are in bold.
> | ANN adapters’ arch | Conv Layers | Params (M) | Spike-ResNet-34 (%) |
> |---------------------|--------------|------------|----------------------|
> | 7                   | 1            | 24.33      | 69.16               |
> | 3                   | 1            | 22.69      | 69.46               |
> | 7-3                 | 2            | 24.77      | **69.82**           |
> | 7-5-3               | 3            | 26.00      | 69.60               |
> | 7-5-3-3-3           | 4            | 26.44      | 66.96               |
> | 7-7-5-5-5-3-3-3-3   | 8            | 31.75      | 55.46               |
>
> #### Table VII. Effect of auxiliary ANN adapters’ number on accuracy and parameters (ImageNet-100, K = 128). Best results are in bold.
> | ANN Adapter Number | Spike-ResNet-34 Acc (%) | MS-ResNet-34 Acc (%) | Spike- and MS-ResNet-34 (Params (M)) | Spike-ResNet-101 Acc (%) | MS-ResNet-101 Acc (%) | Spike- and MS-ResNet-101 (Params (M)) |
> |--------------------|--------------------------|------------------------|------------------------|----------------------------|-------------------------|-------------------------|
> | 1                  | 64.80                   | 70.82                 | 22.61                 | 38.78                     | 70.18                  | 44.10                  |
> | 2                  | 67.42                   | 71.36                 | 23.47                 | 43.20                     | 71.32                  | 45.50                  |
> | 3                  | **69.60**               | **71.88**             | 24.77                 | **59.62**                 | **72.64**              | 46.83                  |
>
> ### 8. Can the distillation heads be pruned after training to reduce inference cost?
> We are sorry for the confusion. Yes. **Intermediate distillation heads are used only during training** and are removed afterward. Inference uses only the student SNN, so cost is unchanged.
>
> ### 9. Does the reconstruction module affect spike sparsity or latency during inference?
> Table VIII compares the accuracy and spiking firing rate of our method with and without the reconstruction module. The reconstruction module is only used during training. During testing, our method incurs no additional cost and behaves the same as other distillation approaches. The results show that the firing rate of our method with the reconstruction module remains comparable to that without the reconstruction module.
> #### Table VIII.   Comparison of firing rate of our method with and without the reconstruction module on ImageNet-1K with MS-ResNet-34.
> | Method | Accuracy | Firing Rate |
> |---------------|----------|-------------|
> | with $L_{rec}$       | 65.95%   | 20.52%      |
> | w/o $L_{rec}$         | 65.56%   | 20.49%      |
> ### 10. How would the method perform if the teacher is another SNN rather than an ANN?
> Thanks for the interesting question. Conceptually, our framework is agnostic to the teacher architecture. One can use an SNN teacher and use its logits in the same multi-stage adapters and local KL losses, and the impedance-driven decomposition would still apply.
>
> However, in this work we choose a strong ANN foundation teacher. The main reason is current SNNs still lag behind SOTA ANNs in accuracy and robustness, and training very deep SNNs remains challenging. Such models can provide significantly richer supervision and benefit mitigating the information-impedance of deep SNNs.
>
> Most practical SNN KD setups in the literature use ANN-SNN distillation, to leverage the richer representations of ANNs to compensate for the optimization and capacity issues of deep SNNs. If the teacher were an SNN of similar or lower capacity, we would expect smaller gains, since the student would be upper-bounded by the SNN teacher.

---

> ### Author Response · Authors · 2025-11-25
>
> ### 11. Can “information impedance” be quantitatively estimated during training to adaptively guide loss weights?
> Thank you for the insightful question. In principle, our notion of “information impedance” could be estimated online, e.g., via reconstruction-based proxies, and used to adaptively tune the stage-wise loss weights. However, in this work, we use fixed weights for two reasons. First, unbiased low-variance estimation of $I(X;Z_l)$ during training is non-trivial and would introduce computational overhead. Second, as shown in Table IX, our ablations show that the proposed multi-stage design is already fairly robust to a wide range of static weights, which can yield significant improvement over prior SOTA without dynamic scheduling (also see Table 1 in the manuscript). We view more sophisticated “impedance-aware” weight adaptation as a promising extension for future work.
> #### Table IX. Effect of depth dependent auxiliary head weights on accuracy (ImageNet-100, Spike-ResNet-101). Best results are in bold.
> | Setting  | $w_c(1)$, $w_k(1)$ | $w_c(2)$, $w_k(2)$ | $w_c(3)$, $w_k(3)$ | Top-1 Acc. (%) |
> |----------|---------------------------|---------------------------|---------------------------|-----------------|
> | Baseline | 1.0                       | 1.0                       | 1.0                       | 69.54           |
> |          | 0.8                       | 1.0                       | 1.2                       | **70.14**           |
> |          | 0.6                       | 1.0                       | 1.4                       | 69.03           |
> |          | 0.4                       | 1.0                       | 1.6                       | 68.64           |
>
> ### 12. Summary
> Once again we thank the reviewer for the detailed and valuable feedback. We hope the above replies can address your concerns.
>
> Our work provides **an information-theoretic view of deep SNN training by introducing information impedance, with SNN-specific entropy bounds and empirical information-flow measurements that reveal severe, depth-accumulated capacity loss in deep SNNs**, especially for fully-spiking SNNs. Motivated by this, we propose a multi-stage distillation framework using a strong ANN foundation teacher, which **decomposes a deep high-impedance path into low-impedance stages, to effectively mitigate the information impedance** caused by spike quantization.
>
> Our method **substantially boosts the learning of deep residual SNNs**, and the gains are particularly significant for fully-spiking SNNs and deeper models. On ImageNet-1K, our method achieves **77.14\%** accuracy with MS-ResNet-101 and **72.86\%** with fully-spiking ResNet-101, surpassing the prior SOTA by **2.93\%** and **4.48\%**, respectively. Our results mark **the first spiking ResNet to surpass the 77\% threshold on ImageNet-1K**. Notably, unlike vanilla KD being sufficient for ANNs on large-scale dataset, our work shows that, for SNNs, overcoming the information impedance is essential to fully unlock the potential of SNN distillation.

---

### Official Review · Reviewer_WDs9 · 2025-10-31

**Soundness:** 2
**Presentation:** 2
**Contribution:** 2
**Rating:** 4
**Confidence:** 4

**Summary:**

This paper presents a novel multi-stage knowledge distillation framework for deep spiking neural networks, supported by an information-theoretic analysis that identifies information impedance as a key bottleneck. The proposed approach leverages supervision from a large teacher model (DINOv2) across multiple intermediate stages to mitigate information loss in deep SNNs. Extensive experiments on ImageNet and its variants demonstrate significant performance improvements over prior methods. The work is well-motivated and clearly written, combining theoretical insight and practical implementation. However, some important aspects—such as energy efficiency, feature alignment, and theoretical validation of the mitigation process—require further clarification.

**Strengths:**

1. This work provides an information-theoretic analysis of SNNs, which identifies increased information impedance as a bottleneck in deep spiking models.

2. This work proposes a multi-stage distillation framework. The multi-stage supervision from the teacher is novel and well-motivated, effectively decomposing a high-impedance path into multiple low-impedance stages.

3. This work evaluates models of different sizes (Spike-ResNet-34/50/101 and their variants), which shows the effectiveness of the multi-stage supervision from the teacher model.

4. The manuscript is clearly written, provides a valuable theoretical and experimental analysis of deep SNNs, and uses a simple modeling approach to improve accuracy.

**Weaknesses:**

1. Energy efficiency and computational cost are not analyzed (only firing rate is reported), although energy savings are a core motivation for using SNNs.

2. The feature alignment between the ANN block and the SNN student is not clearly explained. It is unclear how the spike features are aligned with continuous teacher features.

3. Knowledge distillation using DINOv2 can increase training cost significantly, while the goal of SNNs is to reduce computational cost.

4. Although the theoretical motivation behind multi-stage supervision is shown, its implications could have been further analyzed with information theory (e.g., choice of the number of layers for each stage of supervision).

**Questions:**

1. Authors are suggested to report computational complexity and energy consumption in comparison with existing works.

2. How are the spiking features aligned with the ANN blocks during distillation? Since the features are of different types (spiking and continuous, respectively), the alignment strategy should be mentioned.

3. Apart from the datasets used, the authors may test the method on event-based or neuromorphic datasets (e.g., Spiking Heidelberg Digits and Spiking Speech Command datasets).

4. Authors may provide an ablation study on the number of supervision stages for a fixed-size model.

5. The teacher model size can be varied as an ablation to show the effect of using large or small teacher models.

6. The robustness results (Table 2) have not been compared with previous works; such a comparison may be included.

---

> ### Author Response · Authors · 2025-11-25
>
> We sincerely thank the reviewer for the valuable comments and constructive suggestions. In the following we provide replies to address your concerns.
> ### 1.  Knowledge distillation using DINOv2 can increase training cost significantly, while the goal of SNNs is to reduce computational cost.
> Thank you for the comment. We have conducted additional evaluation to compare the training overhead of direct training, vanilla KD (ANN ResNet teacher of the same architecture), vanilla KD (DINOv2-base), and our method for different SNN ResNet students, as shown in the following Table I. Compared with vanilla KD (DinoV2), **our method introduces only a marginal increase in training time and memory usage**. Using DINOv2-base as the teacher introduces an increase in training time. However, the relative overhead decreases as the network depth grows. We have added this comparison into Section 6.3.
>
> In the revised version, we also added an inference energy consumption analysis (in Appendix A.4 in the revised manuscript, please also see the following Table II). **Considering inference energy consumption, our method achieves energy savings comparable to direct training and standard KD**, since it does not increase the firing rate drastically and introduces no extra spiking computation at inference. Using DINOv2 only increases the training cost.
>
> #### Table I. Training cost comparison under different teacher models on ImageNet1k (batchsize=1024, DINOv2-base denotes the DINOv2-base teacher, ResNet denotes the ANN teacher of the same depth).
> | Methods             | MS-ResNet34 (Time)      | MS-ResNet34 (Mem, GB)  | MS-ResNet50 (Time)      | MS-ResNet50 (Mem, GB)  | MS-ResNet101 (Time)     | MS-ResNet101 (Mem, GB) |
> |---------------------|--------------------|---------------------|---------------------|---------------------|----------------------|----------------------|
> | Direct Train        | 6m52s              | 78.7                | 10m45s              | 167.5               | 14m53s               | 228.1                |
> | Vanilla KD (ResNet) | 7m38s (+11.2%)     | 80.5 (+2.3%)        | 11m29s (+6.8%)      | 169.9 (+1.4%)       | 16m34s (+11.3%)      | 230.3 (+1.0%)        |
> | Vanilla KD (DINOv2) | 9m29s (+38.1%)     | 89.8 (+14.1%)       | 13m41s (+27.2%)     | 173.4 (+3.5%)       | 18m22s (+23.4%)      | 235.4 (+3.2%)        |
> | Our Method (DINOv2) | 10m08s (+47.6%)    | 91.0 (+15.6%)       | 14m23s (+33.7%)     | 176.4 (+5.3%)       | 19m10s (+28.8%)      | 240.4 (+5.4%)        |
>
> #### Table II. Inference energy comparison of MS-ResNet-34 trained with different teachers on ImageNet-1K.
> | Teacher model | Firing rate (%) | Energy (mJ / img) |
> |--------------|-----------------|--------------------|
> | DinoV2-small     | 20.40           | 0.765              |
> | DinoV2-base     | 20.52           | 0.769              |
> | DinoV2-large    | 20.56           | 0.771              |
> | ResNet34     | 20.53           | 0.770              |

---

> ### Author Response · Authors · 2025-11-25
>
> ### 2.  Although the theoretical motivation behind multi-stage supervision is shown, its implications could have been further analyzed with information theory (e.g., choice of the number of layers for each stage of supervision).
> Thanks for the suggestion. We have conducted additional evaluation on the effect of stage separation, as well as varying weights and temperature of different stages. The results are provided in the following Tables III-V. The results show that evenly distributing ANN adapters yields better accuracy, whereas concentrating them in deeper layers leads to degraded performance. Moreover, moderately increasing the KL-related weights improves performance. We have added these results in Appendix in the revised manuscript.
> #### Table III. Effect of stage separation on accuracy (ImageNet-100, Spike-ResNet-101). Best results are in bold.
> | Setting  | position-1 | position-2 | position-3 | Top-1 Acc. (%) |
> |----------|------------|------------|------------|----------------|
> | Baseline | 22         | 49         | 76         | 69.54          |
> |          | 19         | 40         | 61         | **70.06**      |
> |          | 40         | 61         | 79         | 63.42          |
> |          | 22         | 37         | 55         | 69.84          |
> |          | 55         | 67         | 82         | 52.02          |
> #### Table IV. Effect of depth dependent auxiliary head weights on accuracy (ImageNet-100, Spike-ResNet-101). Best results are in bold.
> | Setting  | $w_c(1)$, $w_k(1)$ | $w_c(2)$, $w_k(2)$ | $w_c(3)$, $w_k(3)$ | Top-1 Acc. (%) |
> |----------|---------------------------|---------------------------|---------------------------|-----------------|
> | Baseline | 1.0                       | 1.0                       | 1.0                       | 69.54           |
> |          | 0.8                       | 1.0                       | 1.2                       | **70.14**           |
> |          | 0.6                       | 1.0                       | 1.4                       | 69.03           |
> |          | 0.4                       | 1.0                       | 1.6                       | 68.64           |
> #### Table V. Effect of depth dependent temperature schedules on accuracy (ImageNet-100, Spike-ResNet-101). Best results are in bold.
> | Setting                  | $\tau_1$ | $\tau_2$  |$\tau_3$  | Top-1 Acc. (%) |
> |--------------------------|------------------|------------------|------------------|-----------------|
> | Baseline                 | 5                | 5                | 5                | 69.54           |
> | Deeper heads sharper     | 3              | 5              | 7              |     69.42            |
> | Shallower heads sharper  | 7              | 5               | 3              |          69.66       |
>
> ### 3.  Authors are suggested to report computational complexity and energy consumption in comparison with existing works.
> Thank you for this constructive suggestion. We have evaluated energy consumption and spike firing rate under various distillation methods and architectures on CIFAR100, as summarized in Table VI. **Our method shows lower firing rate and energy consumption**. The energy comparison of MS-ResNet-34 trained with different teachers on ImageNet-1K is given in the above Table II.
> #### Table VI. Energy comparison with other methods on CIFAR-100 using MS-ResNet-18.
> | Methods        | Firing Rate | Energy |
> |----------------|-------------|--------|
> | Vanilla SNN KD | 22.85%      | 0.42mJ |
> | BKDSNN         | 34.93%      | 0.52mJ |
> | TSER           | 22.95%      | 0.42mJ |
> | **Ours**       | 21.25%      | 0.39mJ |
>
> ### 4.  How are the spiking features aligned with the ANN blocks during distillation? Since the features are of different types (spiking and continuous, respectively), the alignment strategy should be mentioned.
> We are sorry for the confusion. **Our method does not perform feature alignment between ANN teacher features and SNN spike features**. Due to the strong heterogeneity between continuous ANN features and discrete spiking representations, we intentionally avoid intermediate feature matching. Instead, each stage uses a lightweight ANN adapter to map SNN spike representations into distillable soft logits, and the distillation is applied only through local KL on teacher posteriors, not through feature-level alignment. Thus, our method aligns probabilistic outputs rather than heterogeneous intermediate features, which yields more significant improvement on deep SNNs.

---

> ### Author Response · Authors · 2025-11-25
>
> ### 5.  Apart from the datasets used, the authors may test the method on event-based or neuromorphic datasets (e.g., Spiking Heidelberg Digits and Spiking Speech Command datasets).
> Thank you for the suggestion. To enable a fair and direct comparison with prior state-of-the-art approaches, we have additionally evaluated our method on the neuromorphic benchmark CIFAR10-DVS. The results have been added to Appendix B.2 in the revised manuscript.
> #### Table VII. Performance comparison of top-1 accuracy (%) on CIFAR10-DVS.
> | Method     | Model     | Timestep | Acc. (%) |
> |------------|-----------|----------|----------|
> | STBP-tdBN  | ResNet-19 | 10       | 67.80    |
> | Dspike     | ResNet-18 | 10       | 75.40    |
> | SM         | ResNet-18 | 10       | 83.19    |
> | FRTD       | ResNet-18 | 4        | 83.50    |
> | FRTD       | ResNet-18 | 10       | 86.40    |
> | **Ours**    | ResNet-18 | 4        | **84.20** |
> | **Ours**    | ResNet-18 | 10       | **86.80** |
>
> ### 6.  Authors may provide an ablation study on the number of supervision stages for a fixed-size model.
> Thank you for your suggestion. The ablation study on the number of supervision stages is provided in Table 9 in the Appendix in our manuscript, which is also provided here in Table VIII.
> #### Table VIII. Effect of auxiliary ANN adapters’ numbers on accuracy and parameters (ImageNet-100, K = 128). Best results are in bold.
> | ANN Adapter Number | ResNet-34 Spike Acc (%) | ResNet-34 MS Acc (%) | ResNet-34 Params (M) | ResNet-101 Spike Acc (%) | ResNet-101 MS Acc (%) | ResNet-101 Params (M) |
> |--------------------|--------------------------|------------------------|------------------------|----------------------------|-------------------------|-------------------------|
> | 1                  | 64.80                   | 70.82                 | 22.61                 | 38.78                     | 70.18                  | 44.10                  |
> | 2                  | 67.42                   | 71.36                 | 23.47                 | 43.20                     | 71.32                  | 45.50                  |
> | 3                  | **69.60**               | **71.88**             | 24.77                 | **59.62**                 | **72.64**              | 46.83                  |
> ### 7. The teacher model size can be varied as an ablation to show the effect of using large or small teacher models. The robustness results (Table 2) have not been compared with previous works; such a comparison may be included.
>
> Thank you for the suggestion. Under the same experimental setting as Table 2 in the manuscript, we further evaluated the performance of different teacher models and methods for comparison. It can be seen that, using a large DinoV2 as the teacher yields better performance than ResNet34 on ImageNet-1k. Meanwhile, it delivers substantially stronger performance on robustness benchmarks such as ImageNet-C and fine-grained recognition tasks, which addresses a gap in prior work that ignored robustness.
> We have added these results into Table 2 in the revised manuscript.
> #### Table IX. Performance of MS-ResNet-34 under different teachers on Imagenet-1k and ImageNet-C (Top-1 accuracy %).
> | Method       | Teacher Params | ImageNet-1k | ImageNet-C (severity 5) | ImageNet-C (severity 1-5) |
> |-------------|--------|---------|---------------|------------------|
> | w/o Teacher | –      | 63.80   | 20.35         | 34.90            |
> | TSER        | 21.8M  | 65.18   | 21.82         | 36.81            |
> | **Our method with different teachers** |        |         |
> | ResNet34    | 21.8M  | 65.33   | 21.66         | 35.62            |
> | DINOv2-Small| 21M    | 65.40   | 22.94         | 37.67            |
> | DINOv2-Base | 86M    | 65.95   | 23.40         | 38.12            |
> | DINOv2-Large| 300M   | 65.89   | 23.93         | 38.84            |
>
> #### Table X. Performance of MS-ResNet-34 on  fine-grained datasets under different teachers (Top-1 accuracy %).
> | Method       | Teacher  Params | Cars  | Birds | Dogs  | Aircraft | Avg   |
> |-------------|--------|-------|-------|-------|----------|-------|
> | w/o Teacher | –      | 42.83 | 42.96 | 67.93 | 42.24    | 48.99 |
> | TSER        | 21.8M  | 42.32 | 42.54 | 68.66 | 41.08    | 48.65 |
> | **Our method with different teachers** |
> | ResNet34    | 21.8M  | 41.71 | 42.61 | 68.76 | 41.21    | 48.57 |
> | DINOv2-Small| 21M    | 42.13 | 44.60 | 70.49 | 42.69    | 49.98 |
> | DINOv2-Base | 86M    | 42.57 | 45.07 | 70.22 | 42.88    | 50.18 |
> | DINOv2-Large| 300M   | 43.12 | 45.49 | 70.92 | 43.58    | 50.78 |

---

### Official Review · Reviewer_U5GF · 2025-10-31

**Soundness:** 3
**Presentation:** 3
**Contribution:** 2
**Rating:** 6
**Confidence:** 5

**Summary:**

The paper studies an **information impedance** bottleneck in deep SNNs: spike quantization & spike-only propagation compress mutual information as depth grows. It formalizes the target and derives layer-wise capacity/bottleneck results under spike-count coding. Building on this, the authors propose multi-stage knowledge distillation (MSKD) from a foundation teacher (DINOv2), adding intermediate heads via lightweight ANN adapters and an input reconstruction loss. On ImageNet-1K, they report strong results for deep spiking ResNets (e.g., MS-ResNet-101 with T=4 reaches 77.14%), with especially large gains for fully-spiking Spike-ResNet-50/101.

**Strengths:**

1) The paper offers a perspective that the training difficulty of deep SNNs to an information bottleneck/impedance and theoretically characterizes layer-wise capacity and its depth scaling, aligning analysis with observed phenomena.

2) Multi-stage distillation decomposes a high-impedance learning paths into low-impedance stages, injecting KL objectives at multiple depths and adding reconstruction to preserve input information; the design matches the motivation.
2)  On ImageNet-1K, MS-ResNet-101 (T=4) reaches 77.14% and surpasses prior SOTA; gains are larger for fully-spiking Spike-ResNet-50/101.
2) The paper compares against vanilla KD, removes the reconstruction term, varies depth/architectures, and reports robustness and teacher-size effects.

**Weaknesses:**

1) **Originality of multi-stage distillation.** While I appreciate the theoretical treatment of information impedance, the multi-stage distillation design appears relatively conventional. Elements such as intermediate heads, local KL terms, and auxiliary reconstruction have prior art in both ANN and certain SNN KD settings. The novelty seems to lie more in the systematic “information impedance” perspective and its integration tailored to deep SNNs. I suggest clarifying how your approach differs from **deep supervision, layer-wise distillation, and prior SNN KD methods**.
2) **Depth-aware objective design.** Given the claim that **information impedance grows with depth**, Eqs. (8–10) could go further by explicitly making the supervision depth-aware. Concretely, (i) increase KL/reconstruction weights with layer depth, (ii) place denser intermediate heads toward later layers, and/or (iii) use a depth-dependent temperature to sharpen targets where the bottleneck is strongest.
3) **Training cost & fully-spike consistency.** The method depends on ANN adapters and a large DINOv2 teacher. Although intermediate heads are removed during inference, please quantify the additional training overhead—specifically in terms of compute (FLOPs or GPU-hours), memory usage, energy consumption.
4) **Missing SEW-ResNet-50/101 results under the proposed distillation.** While Table 1 includes prior distillation baselines for SEW-ResNet-34, it does not report SEW-ResNet-34/50/101 results under the authors’ proposed distillation, and this omission is not explained.

**Questions:**

Please refer to the four points included in **Weaknesses** for details.

---

> ### Author Response · Authors · 2025-11-25
>
> Dear Reviewer,
> we sincerely thank you for the valuable feedback, below we provide replies to address your concerns.
> ### 1. Originality of multi-stage distillation. While I appreciate the theoretical treatment of information impedance, the multi-stage distillation design appears relatively conventional. Elements such as intermediate heads, local KL terms, and auxiliary reconstruction have prior art in both ANN and certain SNN KD settings. The novelty seems to lie more in the systematic “information impedance” perspective and its integration tailored to deep SNNs. I suggest clarifying how your approach differs from deep supervision, layer-wise distillation, and prior SNN KD methods.
>
> Thanks for the suggestion. We have added a discussion in Section 2 (Related Work) to clarify the difference from these existing methods. **While intermediate heads and local KL terms resemble deep supervision or layer-wise distillation in form, our contribution lies not only in architectural design but in a theory-driven motivation and mechanism specifically tailored for deep SNNs, whose effectiveness is strongly and consistently validated by our extensive empirical results on deep SNNs.**
>
> **(1) Comparison with deep supervision.** Deep supervision typically adds intermediate CE with one-hot labels mainly for optimization shortcutting [Ref1]. In contrast, our intermediate heads distill high-entropy teacher posteriors from a foundation model via local KL, **particularly targeting information impedance of deep SNNs.** Importantly, our experiments show that (Table I):
>  - **For ANN, vanilla KD is sufficient and multi-stage KD does not yield any improvement.** This is consistent with previous results that, for ANNs the vanilla distillation method is sufficient on large-scale datasets [Ref2].
>  - **For both fully-spiking and hybrid SNNs, our method achieves significant improvement over the vanilla method.** The improvement is particularly pronounced for the fully-spiking ResNet.
>
> Our method demonstrates particular effectiveness on deeper SNN architectures. For Spike-ResNet-101, which suffers from pronounced information degradation under its fully-spiking design, our method achieves a remarkable improvement of +59.58\% over direct training. Overall, these results clearly demonstrate the effectiveness of our method in alleviating the information transmission bottleneck in deep SNNs.
> #### Table I. Comparison of accuracy (%) on ImageNet100 across different architectures and depths.
> | Method                 | Spike-ResNet-34      | ResNet34 (ANN)      | Spike-ResNet-50      | Spike-ResNet-101      |
> |------------------------|------------------------|---------------------------|------------------------|-------------------------|
> | Baseline               | 71.50                 | 82.58               | 69.85                 | 9.78                   |
> | $𝐿_{kd}$                   | 77.04 (+5.54)         | **87.76 (+5.18)**       | 77.36 (+7.51)         | 15.60 (+5.82)          |
> | w/o $L_{rec}$              | 80.32 (+8.82)         | 87.65 (+5.02)       | 80.26 (+10.41)        | 68.76 (+58.98)         |
> | **Ours**               | **80.81 (+9.31)**     | 87.53 (+4.95)        | **80.86 (+11.01)**    | **69.36 (+59.58)**      |
>
> **(2) Different from layer-wise feature-alignment KD and prior SNN KD.** Most prior SNN KD methods follow ANN-style logit matching or intermediate feature alignment [Ref3, Ref4]. **For SNNs, the performance of intermediate feature matching based KD is limited by severe ANN-SNN feature heterogeneity** (continuous vs. spiking representations). **Our lightweight ANN adapters map spiking features to distillable logits, and we align teacher posteriors without hard intermediate feature matching, which is explicitly tailored to overcome impedance.**
>
> (3) Explicit input-information preservation. We add a reconstruction loss to preserve input information after spike encoding, which is not a standard component of deep supervision or layer-wise KD.
>
> Empirically, as shown in Table 1 in the manuscript, our method significantly outperforms representative prior SNN KD baselines, and the gain is particularly significant on deep fully-spiking models, which matches our analysis. Particularly, **on ImageNet-1k, our method outperforms existing SOTA results by more than 4% on spike-ResNet-101.**
>
> - *[Ref1] C. Lee, et al. Deeply supervised nets. AISTATS, 2015.*
> - *[Ref2] Z. Hao, et al. Revisit the power of vanilla knowledge distillation: from small scale to large scale. NeurIPS, 2023.*
> - *[Ref3] BKDSNN: Enhancing the Performance of Learning-based Spiking Neural Networks Training with Blurred Knowledge Distillation, ECCV, 2024.*
> - *[Ref4] D. Hong, et al. Lasnn: Layer-wise ann-to-snn distillation for effective and efficient training in deep spiking neural networks. arXiv preprint,2023.*

---

> ### Author Response · Authors · 2025-11-25
>
> ### 2. Depth-aware objective design. Given the claim that information impedance grows with depth, Eqs. (8–10) could go further by explicitly making the supervision depth-aware. Concretely, (i) increase KL/reconstruction weights with layer depth, (ii) place denser intermediate heads toward later layers, and/or (iii) use a depth-dependent temperature to sharpen targets where the bottleneck is strongest.
> Thanks for the suggestion. We have revised Eqs. (8)-(10) and conducted evaluation experiments from the three perspectives accordingly. As shown in Tables II–IV, moderately increasing the KL-related weights improves performance. In addition, our results indicate that evenly distributing ANN adapters yields better accuracy, whereas concentrating them in deeper layers leads to degraded performance. We have added these results in Appendix in the revised manuscript.
>
> #### Table II. Effect of depth dependent auxiliary head weights on accuracy (ImageNet-100, Spike-ResNet-101). Best results are in bold.
> | Setting  | $w_c(1)$, $w_k(1)$ | $w_c(2)$, $w_k(2)$ | $w_c(3)$, $w_k(3)$ | Top-1 Acc. (%) |
> |----------|---------------------------|---------------------------|---------------------------|-----------------|
> | Baseline | 1.0                       | 1.0                       | 1.0                       | 69.54           |
> |          | 0.8                       | 1.0                       | 1.2                       | **70.14**           |
> |          | 0.6                       | 1.0                       | 1.4                       | 69.03           |
> |          | 0.4                       | 1.0                       | 1.6                       | 68.64           |
>
> #### Table III. Effect of depth dependent temperature schedules on accuracy (ImageNet-100, Spike-ResNet-101). Best results are in bold.
> | Setting                  | $\tau_1$ | $\tau_2$  |$\tau_3$  | Top-1 Acc. (%) |
> |--------------------------|------------------|------------------|------------------|-----------------|
> | Baseline                 | 5                | 5                | 5                | 69.54           |
> | Deeper heads sharper     | 3              | 5              | 7              |     69.42            |
> | Shallower heads sharper  | 7              | 5               | 3              |          69.66       |
>
> #### Table IV. Effect of placing 3 ANN adapters at different depths on accuracy (ImageNet-100, Spike-ResNet-101). Best results are in bold.
> | Setting  | position-1 | position-2 | position-3 | Top-1 Acc. (%) |
> |----------|------------|------------|------------|----------------|
> | Baseline | 22         | 49         | 76         | 69.54          |
> |          | 19         | 40         | 61         | **70.06**      |
> |          | 40         | 61         | 79         | 63.42          |
> |          | 22         | 37         | 55         | 69.84          |
> |          | 55         | 67         | 82         | 52.02          |

---

> ### Author Response · Authors · 2025-11-25
>
> ### 3.Training Cost & Fully-Spike Consistency. The method depends on ANN adapters and a large DINOv2 teacher. Although intermediate heads are removed during inference, please quantify the additional training overhead—specifically in terms of compute (FLOPs or GPU-hours), memory usage, and energy consumption.
>
> Thank you for your suggestion. We compared the training overhead of direct training, vanilla KD (ANN ResNet teacher of the same architecture), vanilla KD (DINOv2-base), and our method for different SNN ResNet students, as shown in Table V. Compared with vanilla KD (DinoV2), our method with ANN adapters introduces only a marginal increase in training time and memory usage. Using DINOv2-base as the teacher introduces an increase in training time. However, the relative overhead decreases as the network depth grows. We have added these results into Table 4 in the manuscript.
>
> We also conducted an inference energy analysis, as shown the following Table VI. During inference, our method has comparable energy consumption to direct training and standard KD. Our method using DINOv2 only increases the training time and memory.
>
> #### Table V. Training cost comparison under different teacher models on ImageNet1k (batchsize=1024, DINOv2-base denotes the DINOv2-base teacher, ResNet denotes the ANN teacher of the same depth).
> | Methods             | MS-ResNet34 (Time)      | MS-ResNet34 (Mem, GB)  | MS-ResNet50 (Time)      | MS-ResNet50 (Mem, GB)  | MS-ResNet101 (Time)     | MS-ResNet101 (Mem, GB) |
> |---------------------|--------------------|---------------------|---------------------|---------------------|----------------------|----------------------|
> | Direct Train        | 6m52s              | 78.7                | 10m45s              | 167.5               | 14m53s               | 228.1                |
> | Vanilla KD (ResNet) | 7m38s (+11.2%)     | 80.5 (+2.3%)        | 11m29s (+6.8%)      | 169.9 (+1.4%)       | 16m34s (+11.3%)      | 230.3 (+1.0%)        |
> | Vanilla KD (DINOv2) | 9m29s (+38.1%)     | 89.8 (+14.1%)       | 13m41s (+27.2%)     | 173.4 (+3.5%)       | 18m22s (+23.4%)      | 235.4 (+3.2%)        |
> | Our Method (DINOv2) | 10m08s (+47.6%)    | 91.0 (+15.6%)       | 14m23s (+33.7%)     | 176.4 (+5.3%)       | 19m10s (+28.8%)      | 240.4 (+5.4%)        |
>
> #### Table VI. Inference energy comparison of MS-ResNet-34 trained with different teachers on ImageNet-1K.
> | Teacher model | Firing rate (%) | Energy (mJ / img) |
> |--------------|-----------------|--------------------|
> | DinoV2-small     | 20.40           | 0.765              |
> | DinoV2-base     | 20.52           | 0.769              |
> | DinoV2-large    | 20.56           | 0.771              |
> | ResNet34     | 20.53           | 0.770              |
>
> ### 4. Missing SEW-ResNet-50/101 results under the proposed distillation. While Table 1 includes prior distillation baselines for SEW-ResNet-34, it does not report SEW-ResNet-34/50/101 results under the authors’ proposed distillation, and this omission is not explained.
>
> Thank you for this suggestion. We have conducted additional evaluation on SEW-ResNet-34/50/101 with $T=1$ and $T=4$, and added the results into Table 1 in the revised manuscript. Our method can also achieve significant improvement on SEW-ResNet, especially on SEW-ResNet-50/101. *The improvement of SEW-ResNet-101 ($T{=}4$) in the following table is compared with MS-ResNet-104 ($T{=}5$) in [Ref5]*.
> #### Table VII. Performance (accuracy) of SEW-ResNet models with different depths on ImageNet-1K. ($\uparrow$) denotes the improvement over the previous SOTA of SEW-ResNet with the same architecture and time-step, *except SEW-ResNet-101 ($T{=}4$) is compared with MS-ResNet-104 ($T{=}5$) in [Ref5]*.
> | Model          | T = 1  | T = 4                  |
> |----------------|----------------|--------------------------------|
> | SEW-ResNet-34  | 68.05%         | 73.25% **(↑0.09)**             |
> | SEW-ResNet-50  | 69.64%         | 74.58% **(↑0.82)**             |
> | SEW-ResNet-101 | 71.91%         | 76.77% **(↑2.56)**             |
>
> - *[Ref5] Y. Hu, et al. Advancing spiking neural networks toward deep residual learning. IEEE TNNLS, 2024*

---

### Author Response · Authors · 2025-12-03
**Author Final Remarks**

### **3. Clarifications on factual misunderstandings**
We would also like to clarify a few points where some concerns appear to stem from misunderstanding of our method rather than actual limitations of the approach:

**(1) On *“How are the spiking features aligned with the ANN blocks during distillation?”* (Reviewer WDs9).**

Our method **does not perform feature-level alignment** between continuous ANN features and discrete SNN spikes. As we clarified in the rebuttal, **we deliberately avoid matching heterogeneous intermediate representations**. Instead, each stage uses a lightweight ANN adapter to map the local SNN spike state into soft logits, and distillation is applied only via local KL on teacher posteriors. Thus, our approach aligns probabilistic outputs, not intermediate features.

**(2) On *“despite claims about energy-efficient learning, It remains unclear whether the auxiliary ANN modules are implementable in spike-based hardware”* and *"Can the distillation heads be pruned after training to reduce inference cost?"* (Reviewer L8b7).**

The **auxiliary ANN modules are used only during training and are entirely removed at inference**. The final deployed model is a pure SNN, with the same inference architecture and cost as standard SNN distillation methods. Therefore, our claims about energy-efficient spike-based inference do not rely on running any ANN components on neuromorphic hardware, only the SNN runs at test time.

---

### Author Response · Authors · 2025-12-03
**Author Final Remarks**

We sincerely thank the reviewers and the Area Chair for their effort in reviewing our paper.

Our work addresses the core challenge of training deep SNNs by identifying **information impedance**, arising from spiking activations and spike-based propagation, as the fundamental bottleneck. Building on this insight, we introduce a theory-driven solution:
- **An information-theoretic analysis that reveals how spiking activations and spike-based propagation induce increasing information impedance with depth**, fundamentally limiting the learning capacity of deep SNNs.
-  **A multi-stage distillation framework that decomposes a high-impedance learning path into multiple low-impedance stages**, leveraging a strong ANN teacher to enhance information transmission and alleviate spike-induced representational bottlenecks.
- **Significant performance improvements on deep SNNs** on ImageNet-1K, achieving **72.86% with fully-spiking Spike-ResNet-101** and **77.14% with hybrid MS-ResNet-101**, surpassing prior SOTA by **+4.48%** and **+2.93%**, respectively.
### **1. Strengths highlighted by the reviewers**

**(1) Insightful analysis.**
Reviewers appreciated our information-theoretic framing and the **identification of information impedance as a fundamental bottleneck in deep SNNs**:
* (U5GF) remarked that our work *“ties the training difficulty of deep SNNs to an information bottleneck/impedance and characterizes its depth scaling.”*
* (WDs9) highlighted our *“valuable theoretical and experimental analysis of deep SNNs.”*
* (L8b7) noted that the paper *“provides an insightful information-theoretic formulation and a fresh analytical lens for SNN limitations.”*
* (DrHd) praised the *“principled and insightful problem analysis,”* supported by both theoretical propositions and empirical evidence.

**(2) Well-motivated method.**
Reviewers found the proposed multi-stage distillation method is well-motivated and directly addresses the identified bottleneck:
* (U5GF) commented that our design *“decomposes a high-impedance learning path into low-impedance stages, and the design matches the motivation.”*
* (WDs9) commented that the method is *“novel and well-motivated.”*
* (DrHd) described it as a *“well-motivated and effective method... direct and logical solution to the problem of information impedance.”*

**(3) Strong results.**
Our experiments were recognized as demonstrating significant gains across deep SNN architectures:
* (U5GF) highlighted the strong ImageNet results: *“MS-ResNet-101 reaches 77.14%, with especially large gains for fully-spiking Spike-ResNet models.”*
* (WDs9) noted that *“extensive experiments demonstrate significant performance improvements over prior methods.”*
* (L8b7) pointed out the *“significant gains on ImageNet-1K.”*
* (DrHd) concluded that our method *“achieves a new state-of-the-art accuracy on ImageNet, marking an advance for the field.”*
### **2. Our Rebuttal to Reviewers' Concerns**
We provided thorough responses to address the concerns of the reviewers:
- Clarified clearly the difference from existing multi-stage methods in ANNs and intermediate feature alignment methods.
- Added training overhead comparison with direct training, vanilla KD, and our method for different student models and different teachers, showing that compared with vanilla KD, **our method with ANN adapters introduces only a marginal increase in training time and memory usage**.
- Provided energy consumption analyses to show that our method has **comparable inference energy consumption to direct training and standard KD**.
- Added evaluation with different teacher models (ResNet34, DINOv2-S/B/L) on ImageNet-1K, ImageNetC and Fine-Grained datasets, showing that using **a large DinoV2 as the teacher yields better accuracy and stronger robustness**.
- Added experiments on neuromorphic dataset CIFAR10-DVS to demonstrate the applicability of our framework to neuromorphic datasets.
- Clarified the reconstruction module design, detailing its architecture and clarifying that it does not increase inference energy and latency.
- Expanded depth dependent ablations, covering positions of auxiliary ANN modules, KL-weight schedules, and temperature variations to validate our design choices.
- Added evaluation to demonstrate the effectiveness of our method on SEW-ResNet-50/101.

While the reviewers have not yet replied to our responses, we believe that our clarifications and added evaluations can adequately address the reviewers' concerns.

---

### Meta-Review · Area_Chair_jtNe · 2026-01-06

**Summary:**

This paper investigates the difficulty of training deep spiking neural networks (SNNs) through an information-theoretic lens, attributing the degradation of performance with depth to an “information impedance” induced by spiking quantization and spike-based propagation. Motivated by this analysis, the authors propose a multi-stage knowledge distillation framework that injects supervision from a high-capacity ANN teacher (DINOv2) at multiple intermediate depths using auxiliary ANN adapters. Extensive experiments on ImageNet-1K and related benchmarks show large empirical gains for deep and fully-spiking ResNet-style SNNs.

Reviewers broadly agree that the paper is well written, carefully engineered, and empirically strong, with particularly impressive ImageNet results for deep SNNs. However, the discussion did not converge on acceptance due to fundamental concerns regarding methodological novelty, conceptual framing, and scientific positioning. Despite substantial rebuttal effort and many additional experiments, these core issues remain unresolved.

**Reviewer Concerns:**

Concerns that were largely addressed

Several reviewers’ requests for additional ablations, training-cost analysis, energy estimates, robustness evaluation, and neuromorphic datasets were thoroughly addressed in the rebuttal.

The authors clarified that auxiliary ANN adapters and reconstruction modules are training-only, alleviating concerns about inference-time energy and latency.

Empirical validation was significantly strengthened, including deeper models, additional teachers, SEW-ResNet results, CIFAR10-DVS, and detailed overhead analyses.

Theoretical exposition and connection to information-theoretic concepts were expanded and clarified in response to reviewer feedback.

Outstanding concerns that remain

Limited methodological novelty: Multiple reviewers (notably L8b7 and U5GF) raised persistent concerns that the proposed multi-stage distillation framework largely recombines well-established components—deep supervision, auxiliary heads, and knowledge distillation—without introducing a fundamentally new algorithmic mechanism. While the “information impedance” framing is insightful, it is widely viewed as a reinterpretation or specialization of known information bottleneck arguments, rather than a new theoretical advance.

Conceptual contribution vs. engineering refinement: The strongest contribution of the paper lies in its analysis and synthesis, rather than in the distillation method itself. Several reviewers explicitly noted that the novelty is primarily in the interpretation and justification, not in the method’s structure, making the contribution feel incremental relative to prior ANN and SNN KD literature.

Dependence on large foundation teachers: The reliance on DINOv2 as a teacher raises concerns about generality and alignment with neuromorphic motivations. Although the authors convincingly show training-only overhead, the method’s effectiveness appears tightly coupled to access to very large pretrained models, which limits broader applicability.

Polarized reviewer opinions: While some reviewers rated the paper slightly above threshold, at least one reviewer (L8b7) maintained a firm reject stance with high confidence, and several others explicitly stated they “would not mind if the paper is rejected,” indicating weak support for acceptance.

**Reviewer Scores:**

Reviewer L8b7: Would maintain a reject (2), citing lack of novelty, incremental theoretical contribution, and overstatement of impact.

Reviewer U5GF: Likely to remain borderline accept (6) but explicitly indicated comfort with rejection and raised originality concerns.

Reviewer WDs9: Likely to remain borderline or below threshold (4), with concerns about contribution and framing.

Reviewer DrHd: Likely to remain weak accept (6), but acknowledged that core methodological components are not novel.

Overall, the scores remain borderline and polarized, without a clear consensus in favor of acceptance.

---

### Decision · Program_Chairs · 2026-01-26

Reject